# Why Policy Gradient Algorithms Work for Undiscounted Total-Reward MDPs

## Abstract

The classical policy gradient method is the theoretical and conceptual foundation of modern policy-based reinforcement learning (RL) algorithms. Most rigorous analyses of such methods, particularly those establishing convergence guarantees, assume a discount factor $\gamma < 1$. In contrast, however, a recent line of work on policy-based RL for large language models uses the undiscounted total-reward setting with $\gamma = 1$, rendering much of the existing theory inapplicable. In this paper, we provide analyses of the policy gradient method for undiscounted expected total-reward infinite-horizon MDPs based on two key insights: (i) the classification of the MDP states into recurrent and transient states is invariant over the set of policies that assign strictly positive probability to every action (as is typical in deep RL models employing a softmax output layer) and (ii) the classical state visitation measure (which may be ill-defined when $\gamma = 1$) can be replaced with a new object that we call the transient visitation measure.

## 1 Introduction

Since the seminal Policy Gradient Theorem (Sutton et al., 1999), policy gradient algorithms have been a cornerstone of modern reinforcement learning (RL). Unlike classical dynamic programming approaches, policy gradient methods directly optimize the policy using the gradient of the expected total-reward. These methods, along with their deep learning variants, have achieved remarkable practical success, and their convergence properties have been extensively studied for Markov decision processes (MDP) with discount factor $\gamma < 1$.

More recently, however, a large body of work has emerged on training large language models within the RL framework without discounting ($\gamma = 1$) and arbitrarily long horizons, as in reinforcement learning from human feedback (RLHF) (Christiano et al., 2017) and reinforcement learning with verifiable rewards (RLVR) (Guo et al., 2025). Yet, the convergence of policy gradient methods in this undiscounted total-reward setup remains largely unexplored, and even the policy gradient theorem itself has not been rigorously established in this setup.

**Contribution** In this work, we study the convergence of policy gradient methods for undiscounted expected total-reward infinite-horizon MDPs. Our analysis is based on two key insights: (i) the classification of the MDP states into recurrent and transient states is invariant over the set of policies that assign strictly positive probability to every action (as is typical in deep RL models employing a softmax output layer) and (ii) the classical state visitation measure (which may be ill-defined when $\gamma = 1$) can be replaced with a new object that we call the *transient visitation measure*. Leveraging these insights, we establish convergence guarantees for projected policy gradient and natural policy gradient algorithms in the tabular setting.

### 1.1 Related works

**Undiscounted total-reward infinite horizon MDP.** The setup of undiscounted total-reward MDP was first introduced by Savage (1965). For the well-definedness of the value function, Schäl (1983) considered the finiteness of $V_+^\pi$ and $V_-^\pi$ (defined in the next section) and the existence of an optimal policy was first proved by Van Der Wal (1981). With different additional assumptions, three models of total-reward setup have been proposed and studied: stochastic shortest path model (Eaton &

Zadeh, 1962), positive model (Blackwell, 1967), and negative model (Strauch, 1966). The stochastic shortest path model assumes single absorbing terminal state and the existence of a policy that reaches the terminal state with probability 1 from any initial state (Bertsekas & Tsitsiklis, 1991). The positive model assumed $V_+^\pi$ is finite, and for each $s$, there exist $a$ with $r(s,a) \geq 0$ (Puterman, 2014, Section 7.2). The negative model assumed $V_+^\pi = 0$ and there exist $\pi$ for which $V_-^\pi(s) > \infty$ for all $s$ (Puterman, 2014, Section 7.3). In this work, the undiscounted total-reward model with the assumption that $V^\pi$ is finite, motivated by modern reinforcement learning frameworks and needed to establish the transient policy gradient, does not fall into previous categories. Our setup is also distinct from the average-reward setup, which considers averaging the sum of rewards, while ours do not as clarified in Section 2.

**Policy gradient method.** Policy gradient methods (Williams, 1992; Sutton et al., 1999; Konda & Tsitsiklis, 1999; Kakade, 2001) are foundational reinforcement learning algorithms, commonly implemented with deep neural networks for policy parameterization (Schulman et al., 2015; 2017). In line with their practical success, convergence of policy gradient variants has been extensively studied across settings. In discounted total-reward infinite horizon MDP, Agarwal et al. (2021); Xiao (2022); Bhandari & Russo (2024); Mei et al. (2020) analyzed convergence of projected policy gradient and naive policy gradient with softmax parametrization. The natural policy gradient, introduced by Kakade (2001) and viewable as a special case of mirror descent (Shani et al., 2020), has been analyzed by Agarwal et al. (2021); Cen et al. (2022); Xiao (2022); Lan (2023). In the average reward MDP, convergence results have been established by Even-Dar et al. (2009); Murthy & Srikant (2023); Bai et al. (2024); Kumar et al. (2024), and related analyses exist for the finite horizon setup as well (Hambly et al., 2021; Guo et al., 2022; Klein et al., 2023).

In undiscounted total-reward infinite-horizon MDP, however, there are few results on policy gradient methods. Since Sutton et al. (1999) established the policy gradient theorem only for the discounted total-reward and average reward MDPs, Bojun (2020); Ribera Borrell et al. (2025) analyze policy gradients for the undiscounted total-reward random time horizon MDP. Specifically, Bojun (2020) consider an episodic learning process that can be viewed as an ergodic Markov chain with finite episode length , and establish a policy gradient theorem via the steady state distribution. Ribera Borrell et al. (2025) consider trajectory dependent random termination times and prove a policy gradient theorem with an almost surely finite termination time. We note that neither work further analyzes the convergence of policy gradient methods, and their setups and assumptions differ from ours, as shown in next section.

## 2 UNDISCOUNTED EXPECTED TOTAL-REWARD INFINITE-HORIZON MDPs

In this work, we consider undiscounted total-reward infinite-horizon Markov decision processes (MDPs). We review basic definitions and assumptions of undiscounted MDPs and reinforcement learning (RL). For further details, we refer the readers to references such as (Puterman, 2014, Section 7) or (Sutton & Barto, 2018).

**Undiscounted Markov decision processes.** Let $\mathcal{M}(\mathcal{X})$ be the space of probability distributions over a set $\mathcal{X}$. Write $(\mathcal{S}, \mathcal{A}, P, r, \mu)$ to denote the infinite-horizon undiscounted MDP with finite state space $\mathcal{S}$, finite action space $\mathcal{A}$, transition matrix $P \colon \mathcal{S} \times \mathcal{A} \to \mathcal{M}(\mathcal{S})$, bounded reward $r \colon \mathcal{S} \times \mathcal{A} \to [-R, R]$ with some $R < \infty$, and initial state distribution $\mu \in \mathcal{M}(\mathcal{S})$. We say the reward is nonnegative if $r(s,a) \geq 0$ for all $s \in \mathcal{S}, a \in \mathcal{A}$. Denote $\pi \colon \mathcal{S} \to \mathcal{M}(\mathcal{A})$ for a policy. Define

$$\Pi = \text{set of all policies} = \mathcal{M}(\mathcal{A})^\mathcal{S},$$
$$\Pi_+ = \{\pi \in \Pi \mid \pi(a \mid s) > 0 \text{ for all } s, a\}.$$

So, $\Pi_+$ is the (relative) interior of $\Pi$. Let

$$V_+^\pi(s) = \lim_{T \to \infty} \mathbb{E}_\pi \left[ \sum_{i=0}^{T-1} \max\{r(s_i, a_i), 0\} \,\Big|\, s_0 = s \right]$$

$$V_-^\pi(s) = \lim_{T \to \infty} \mathbb{E}_\pi \left[ \sum_{i=0}^{T-1} \max\{-r(s_i, a_i), 0\} \,\Big|\, s_0 = s \right],$$

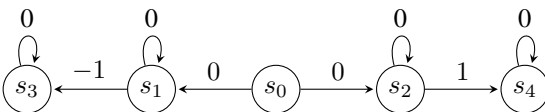

Figure 1: Pathological MDP: The value function is discontinuous at the optimal policy, and the optimal action-value function does not specify the optimal policy

where $\mathbb{E}_\pi$ denotes expectation over trajectories $(s_0, a_0, s_1, a_1, \ldots, s_{T-1}, a_{T-1})$ induced by the policy $\pi$. Since both summands are nonnegative, the monotone convergence theorem guarantees that each limit exists (possibly infinite). To ensure the well-definedness of the value function $V^\pi$, we impose the following assumption.

**Assumption 1** (Finiteness of value function). $V_+^\pi(s) < \infty$, $V_-^\pi(s) < \infty$ *for all* $\pi \in \Pi$ *and* $s \in \mathcal{S}$.

As noted, in recent training of large language models under the frameworks of reinforcement learning with human feedback (RLHF) and reinforcement learning with verifiable reward (RLVR), finite rewards are assigned once at the end of the trajectory, and Assumption 1 holds.

Under Assumption 1, we define the state value function as

$$V^\pi = V_+^\pi - V_-^\pi,$$

and $V_\mu^\pi = \mathbb{E}_{s \sim \mu}[V^\pi(s)]$ where $\mu \in \mathcal{M}(\mathcal{S})$ is the initial state distribution. Likewise, define

$$Q_+^\pi(s, a) = \lim_{T \to \infty} \mathbb{E}_\pi\left[\sum_{i=0}^{T-1} \max\{r(s_i, a_i), 0\} \,\Big|\, s_0, a_0 = s, a\right],$$

and $Q_-^\pi$ analogously. Likewise, define $Q^\pi = Q_+^\pi - Q_-^\pi$. Then, $Q^\pi$ is well-defined under Assumption 1, and $Q^\pi = PV^\pi + r$ where $P \in \mathbb{R}^{|\mathcal{S}||\mathcal{A}| \times |\mathcal{S}|}$, and $V^\pi(s) = \mathbb{E}_{a \sim \pi(\cdot \mid s)}[Q^\pi(s, a)]$ by definition.

We say $V^\star$ is optimal value function if $V^\star(s) = \max_\pi V^\pi(s)$ for all $s \in \mathcal{S}$ and $\pi$ is an $\epsilon$-optimal policy if $\|V^\star - V^\pi\|_\infty \leq \epsilon$. It is known that the optimal value function and an optimal policy always exist in the undiscounted total-reward setup with finite state and action spaces (Puterman, 2014, Theorem 7.1.9). (As a technical detail, Theorem 7.1.9 of Puterman (2014) is stated in terms of the set of all history-dependent policies, but the proof also works for the set of all stationary policies, which is our focus in this work.)

For notational conciseness, we write $r^\pi(s) = \mathbb{E}_{a \sim \pi(\cdot \mid s)}[r(s, a)]$ for the reward induced by policy $\pi$ and $P^\pi(s, s')$ defined as

$$P^\pi(s, s') = \mathrm{Prob}(s \to s' \mid a \sim \pi(\cdot \mid s), s' \sim P(\cdot \mid s, a))$$

is the transition probability induced by policy $\pi$. Then, we can write $V^\pi = \sum_{n=0}^\infty (P^\pi)^n r^\pi$.

In Section 3, we discuss the continuity of the mapping $\pi \mapsto V^\pi$. Since $|\mathcal{S}|$ and $|\mathcal{A}|$ are finite, we can identify $\pi$ and $V^\pi$ as finite-dimensional vectors, namely, as $\pi \in \mathbb{R}^{|\mathcal{S}| \times |\mathcal{A}|}$ and $V^\pi \in \mathbb{R}^{|\mathcal{S}|}$. Therefore, continuity of $\pi \mapsto V^\pi$ can be interpreted as continuity of the mapping from $\mathbb{R}^{|\mathcal{S}| \times |\mathcal{A}|}$ to $\mathbb{R}^{|\mathcal{S}|}$ under the usual metric.

## 3 TROUBLES WITH UNDISCOUNTED TOTAL-REWARD MDPS

In this section, we point out two pathologies that can arise in total-reward MDPs that do not arise in the discounted setup.

**Pathology 1.** *The value function $V^\pi$ may be a discontinuous function of $\pi \in \Pi$, even when $|\mathcal{S}|$ and $|\mathcal{A}|$ are finite and $V^\pi$ is finite.*

In the example of Figure 1, the optimal action at state $s_1$ is to remain at $s_1$. Under the optimal policy, taking this optimal action, we have $V^\star(s_1) = 0$, but any policy assigning a non-zero probability to the other action, transitioning to $s_3$, yields $V^\pi(s_1) = -1$. This example illustrates that the value function can be discontinuous in $\pi$, and a policy gradient method cannot be expected to succeed in the presence of such discontinuities, while the discounted-reward setup guarantees differentiability of the value function. We provide the framework to address this pathology in Section 4.

**Pathology 2.** *The optimal action-value function $Q^\star$ does not, by itself, specify the optimal policy $\pi^\star$. In particular, a policy $\pi$ satisfying $\pi(s) \in \arg\max_{a \in \mathcal{A}} Q^\star(s, a)$ for all $s \in \mathcal{S}$ may not be optimal.*

Again, in the example of Figure 1, the optimal action at $s_2$ is to transition to $s_4$. However, the non-optimal policy $\pi$ that stays at $s_2$ with probability one also satisfies $Q^\star(s_2, \pi(s_2)) = V^\star(s_2) = +1$. In other words, the policy $\pi(s) \in \arg\max_{a \in \mathcal{A}} Q^\star(s, a)$ can be non-optimal, and it is known that additional conditions are needed to specify an optimal policy in this setup (Puterman, 2014, Theorem 7.25). This example illustrates that value-based methods such as value iteration or Q-learning may fail to provide an optimal policy even if they approximate the optimal $Q$-function well. In this work, we study the policy gradient method, a policy-based RL method, and show that it is not subject to this issue.

## 4 RECURRENT-TRANSIENT THEORY OF POLICY GRADIENTS

In this section, we apply the recurrent-transient theory of Markov chains to undiscounted total-reward MDPs, introduce a new object that we term the *transient visitation measure*, and establish a policy gradient theorem.

### 4.1 RECURRENT-TRANSIENT CLASSIFICATION OF STATES

**Definition.** *Given a policy $\pi \in \Pi$, a state $s \in \mathcal{S}$ is recurrent if its return time starting from $s$ is finite with probability 1. Otherwise, $s$ is transient.*

Equivalently, if $n_s$ is the random variable representing the number of visits to state $s$ starting from $s$, then $s$ is recurrent if and only if $\mathbb{E}_\pi[n_s] = \sum_{k=0}^\infty (P^\pi)^k(s, s) = \infty$, and otherwise it is transient (Brémaud, 2013, Theorem 3.1.3).

Let $\pi \in \Pi$. For a given $P^\pi$, the states can be classified into recurrent and transient states, and the Markov chain can be canonically represented as follows (Brémaud, 2013, Section 3.1.3):

$$P^\pi = \begin{bmatrix} \bar{R}^\pi & 0 \\ \bar{S}^\pi & \bar{T}^\pi \end{bmatrix}, \qquad (P^\pi)^n = \begin{bmatrix} (\bar{R}^\pi)^n & 0 \\ \bar{S}_n^\pi & (\bar{T}^\pi)^n \end{bmatrix},$$

where $\bar{R}^\pi$, $\bar{T}^\pi$, and $\bar{S}^\pi$ represent transition probabilities among the recurrent states, among the transient states, and from transient to recurrent states, respectively. This canonical representation exists for any $\pi \in \Pi$, but the recurrent-transient classification of states and the corresponding canonical decomposition of $P^\pi$ may vary.

However, as the following proposition shows, the recurrent-transient classification remains invariant for all $\pi \in \Pi_+$. Recall that $\Pi_+ \subset \Pi$ denotes the set of policies that assign strictly positive probability to every action.

**Proposition 1.** *The recurrent-transient classification of the states does not depend on the choice of $\pi \in \Pi_+$.*

We provide further clarification. For any $\pi \in \Pi_+$, the recurrent-transient classification is determined by the transition kernel $P$, not on the particular choice of $\pi \in \Pi_+$. While a policy $\pi \in \Pi \setminus \Pi_+$ (a policy that assigns zero probability to some actions) may induce a different classification, the algorithms we consider (as well as deep RL algorithms employing a softmax output layer) should be viewed as searching over $\Pi_+$ rather than the full set $\Pi$.

The canonical recurrent-transient decomposition provides the foundation for our analysis of the undiscounted expected total-reward setting. One key consequence of this classification is that the reward at any recurrent state must be zero if the value functions are finite.

**Lemma 1.** *Under Assumption 1 (finiteness of value function), for any $\pi \in \Pi$, if $s$ is a recurrent state, then $r^\pi(s) = 0$.*

Now, define the *transient matrix*:

$$T^\pi = \begin{bmatrix} 0 & 0 \\ 0 & \bar{T}^\pi \end{bmatrix}, \qquad \text{i.e.,} \qquad T^\pi(s_1, s_2) = \begin{cases} P^\pi(s_1, s_2) & \text{if } s_1, s_2 \text{ are both transient} \\ 0 & \text{otherwise.} \end{cases}$$

The transient matrix $T^\pi$ is known to have the following spectral property:

**Fact 1.** *(Berman & Plemmons, 1994, Lemma 8.3.20) Spectral radius of $T^\pi$ is strictly less than $1$.*

This is an important consequence of the recurrent-transient decomposition because the full probability matrix $P^\pi$ will necessarily have a unit spectral radius, and we will use this condition to argue certain convergence results.

By Lemma 1, we have $(P^\pi)^i r^\pi = (T^\pi)^i r^\pi$ for $i \in \mathbb{N}$, which implies

$$V^\pi = \sum_{i=0}^{\infty} (P^\pi)^i r^\pi = \sum_{i=0}^{\infty} (T^\pi)^i r^\pi.$$

By Fact 1 and the classical Neumann series argument, we have $(I - T^\pi)^{-1} = \sum_{i=0}^{\infty} (T^\pi)^i$. These lead the following reformulation of the value function.

**Lemma 2.** *Under Assumption 1, $V^\pi = (I - T^\pi)^{-1} r^\pi$ for any $\pi \in \Pi$.*

### 4.2 Continuity of $V^\pi$ on $\Pi_+$

Returning to the pathological MDP of Figure 1, we observe that (1) at the discontinuous policy, the recurrent-transient classification of states changes, and (2) this transition occurs only on $\Pi \setminus \Pi_+$. Based on these observations and Proposition 1, we obtain the following continuity property of $V^\pi$.

**Lemma 3.** *Under Assumption 1, the mappings $\pi \mapsto V^\pi$ and $\pi \mapsto V_\mu^\pi$ are continuous on $\Pi_+$ for a given $\mu$.*

(Recall $V_\mu^\pi = \mathbb{E}_{s \sim \mu}[V^\pi(s)]$.) In other words, the discontinuity described in Pathology 1 can arise only on the boundary of $\Pi$. Consequently, in our policy gradient methods, we restrict the search to policies $\pi \in \Pi_+$.

Next, define
$$V_+^\star = \sup_{\pi \in \Pi_+} V^\pi \quad \text{and} \quad V_{+,\mu}^\star = \mathbb{E}_{s \sim \mu}[V_+^\star(s)],$$
while recalling
$$V^\star = \max_{\pi \in \Pi} V^\pi \quad \text{and} \quad V_\mu^\star = \mathbb{E}_{s \sim \mu}[V^\star(s)].$$

By definition, $V^\star(s) \geq V_+^\star(s)$ for all $s \in \mathcal{S}$ and $V_\mu^\star \geq V_{+,\mu}^\star$ for any initial distribution $\mu$. Since our policy gradient methods search over $\Pi_+$, they should be thought of as optimizing for $V_{+,\mu}^\star$. However, the distinction between $V_{+,\mu}^\star$ and $V_\mu^\star$ disappears when rewards are nonnegative.

**Lemma 4.** *Under Assumption 1, if rewards are nonnegative, then the map $\pi \mapsto V_\mu^\pi$ are continuous at optimal policies, and $V_{+,\mu}^\star = V_\mu^\star$ for a given $\mu$.*

Under this continuity, we can further show that policy gradient algorithms converge to $V_\mu^\star$.

### 4.3 Transient visitation measure

Conventionally, the state visitation measure is defined as $d_{s_0}^\pi(s) = (1 - \gamma) \sum_{i=0}^{\infty} \text{Prob}(s_i = s \mid s_0; P^\pi)$ in the discounted infinite-horizon setting with $\gamma < 1$. In the undiscounted setting with $\gamma = 1$, this object becomes undefined.

Instead, we define the *transient visitation measure* in the undiscounted total-reward setting using the transient matrix $T^\pi$ as follows:

$$\delta_{s_0}^\pi(s) = \sum_{i=0}^{\infty} \text{Prob}(s_i = s \mid s_0; T^\pi) = e_{s_0}^\intercal (I - T^\pi)^{-1} e_s,$$

where $e_s$ is the $s$-coordinate vector. Also define $\delta_\mu^\pi = \mathbb{E}_{s_0 \sim \mu}[\delta_{s_0}^\pi]$ for an initial state distribution $\mu$. Note that this transient visitation measure is not a probability measure, and, importantly, $\max_{s,s_0 \in \mathcal{S}} \delta_{s_0}^\pi(s) < \infty$ by Fact 1. The transient visitation measure only considers transitions between transient states since these are sufficient to compute the value function, as shown in Lemma 2.

With the transient visitation measure, we can obtain a performance difference lemma in the undiscounted total-reward setup, which will be crucially used in the analysis of policy gradient algorithms.

**Lemma 5** (Transient performance difference lemma). *Under Assumption 1, for $\pi, \pi' \in \Pi$ and a given $\mu$, if $V^{\pi'}(s) = 0$ for all recurrent states $s$ of $P^\pi$, then*

$$V_\mu^\pi - V_\mu^{\pi'} = \sum_{s' \in \mathcal{S}} \sum_{a \in \mathcal{A}} Q^{\pi'}(s', a)\big(\pi(a \mid s') - \pi'(a \mid s')\big)\delta_\mu^\pi(s').$$

### 4.4 TRANSIENT POLICY GRADIENT

We are now ready to present the policy gradient theorem in the undiscounted total-reward setup. Consider the optimization problem

$$\max_{\theta \in \Theta} V_\mu^{\pi_\theta},$$

where $\{\pi_\theta \mid \theta \in \Theta \subset \mathbb{R}^d\}$ is a set of differentiable parametric policies with respect to $\theta$. Based on previous machinery, we establish the following policy gradient theorem.

**Theorem 1** (Transient policy gradient). *Under Assumption 1, for $\pi_\theta \in \Pi_+$,*

$$\nabla_\theta V_\mu^{\pi_\theta} = \sum_{s \in \mathcal{S}} \delta_\mu^{\pi_\theta}(s) \sum_{a \in \mathcal{A}} \nabla_\theta \pi_\theta(a \mid s)Q^{\pi_\theta}(s, a) = \sum_{s \in \mathcal{S}} \delta_\mu^{\pi_\theta}(s) \mathop{\mathbb{E}}_{a \sim \pi_\theta(\cdot \mid s)}[\nabla_\theta \log \pi_\theta(a \mid s)Q^{\pi_\theta}(s, a)].$$

In the following sections, we use this transient policy gradient to analyze the projected policy gradient and natural policy gradient algorithms.

## 5 CONVERGENCE OF PROJECTED POLICY GRADIENT

In this section, we study the convergence of the projected policy gradient algorithm with direct parameterization:

$$\pi_\theta(a \mid s) = \theta_{s,a},$$

where $\theta \in \mathbb{R}^{|\mathcal{S}| \times |\mathcal{A}|}$ satisfying $\sum_{a \in \mathcal{A}} \theta_{s,a} = 1$ and $\theta_{s,a} \geq 0$ for all $s \in \mathcal{S}, a \in \mathcal{A}$. With this direct parameterization, we do not distinguish between the policy $\pi_\theta$ and the parameter $\theta$, and we use $\pi_k$ to denote the iterates of the algorithm.

When using a direct parametrization, we require a mechanism to ensure $\theta$ remains nonnegative and normalized throughout the algorithm. So, we consider the *projected* policy gradient:

$$\pi_{k+1} = \mathbf{proj}_C \big(\pi_k + \eta_k \nabla V_\mu^{\pi_k}\big) \qquad \text{for } k = 0, 1, \ldots,$$

where $C$ is a nonempty closed convex subset of $\Pi$. Usually, $C = \Pi$. But in the undiscounted total-reward setup, we must avoid the (relative) boundary of $\Pi$ as the value function may be discontinuous there, so we consider the following $\alpha$-shrunken $\Pi$ so that the policy set:

$$\Pi_\alpha = \{\pi \mid \pi(s \mid a) \geq \alpha \text{ for all } s \in \mathcal{S}, \ a \in \mathcal{A}\}$$

with $\alpha \in (0, 1)$. For evaluating $\nabla V_\mu^{\pi_k}$, Theorem 1 applied to the direct parametrization setup yields $\nabla V_\mu^{\pi_k}(s, a) = \delta_\mu^{\pi_\theta}(s)Q^{\pi_\theta}(s, a)$ for $\pi \in \Pi_\alpha$.

Normally, the convergence analysis of projected policy gradient requires smoothness (Lipschitz continuity of the gradient) of the value function. For that, we define

$$\max_{\pi \in \Pi_\alpha} \big\|(I - T^\pi)^{-1}\big\|_\infty = C_\alpha < \infty.$$

Note that the mapping $\pi \mapsto P^\pi$ is continuous, and thus mapping $\pi \mapsto T^\pi$ is continuous on $\Pi_+$ since $T^\pi$ can be viewed as the projection of $P^\pi$ onto the transient class, which it is fixed by Proposition 1. Therefore, $C_\alpha$ is finite since $\Pi_\alpha$ is compact and $\big\|(I - T^\pi)^{-1}\big\|_\infty$ is continuous with respect to $\pi$.

**Lemma 6** (Smoothness of value function). *Under Assumption 1, for $\pi, \pi' \in \Pi_\alpha$,*

$$\|\nabla V_\mu^\pi - \nabla V_\mu^{\pi'}\|_2 \leq 2RC_\alpha^2(C_\alpha + 1)|\mathcal{A}|\|\pi - \pi'\|_2.$$

Define $V^{\pi_\alpha^\star} = \max_{\pi \in \Pi_\alpha} V^\pi$ and $V_\mu^{\pi_\alpha^\star} = \mathbb{E}_{s \sim \mu}[V^{\pi_\alpha^\star}(s)]$. Using the Lemma 6, we obtain the following convergence result of the projected policy gradient algorithm.

**Theorem 2.** *Under Assumption 1, for $\alpha \in (0,1)$, $\pi_0 \in \Pi_\alpha$, and given $\mu$ with full support, the projected policy gradient algorithm with step size $\eta = \dfrac{1}{2RC_\alpha^2(C_\alpha+1)|\mathcal{A}|}$ generates a sequence of policies $\{\pi_k\}_{k=1}^\infty$ satisfying*

$$V_\mu^{\pi_\alpha^\star} - V_\mu^{\pi_k} \leq \frac{256R|\mathcal{S}|\,|\mathcal{A}|C_\alpha^2(C_\alpha+1)}{k}\left\|\frac{\delta_\mu^{\pi_\alpha^\star}}{\mu}\right\|_\infty^2.$$

We defer the proofs to Appendix B, but we quickly note that the proof strategy closely follows Xiao (2022), which considers the discounted reward setup and uses the (non-transient) visitation measure.

Theorem 2 shows that $V_\mu^{\pi_k} \to V_\mu^{\pi_\alpha^\star}$ with a sublinear rate, and since $V_\mu^{\pi_\alpha^\star} \to V_{+,\mu}^\star$ as $\alpha \to 0$, we can define an iteration complexity for finding an $\epsilon$-optimal policy by projecting onto $\Pi_\alpha$ with the value of $\alpha$ chosen to be a function of $\epsilon$ with nonnegative reward.

**Corollary 1.** *Assume the rewards are nonnegative. For any given $\epsilon \in (0,1)$ and $\mu$ with full support, set $\alpha = \dfrac{\epsilon}{2|\mathcal{S}||\mathcal{A}|\left\|\delta_\mu^{\pi^\star}\right\|_\infty\|Q^{\pi^\star}\|_\infty}$ and let the step size be $\eta = \dfrac{1}{2RC_\alpha^2(C_\alpha+1)|\mathcal{A}|}$. Then, under Assumption 1, for $\pi_0 \in \Pi_\alpha$, the iterates of projected policy gradient $\pi_k$ are $\epsilon$-optimal policy for*

$$k \geq 512(1/\epsilon)R|\mathcal{S}|\,|\mathcal{A}|C_\alpha^2(C_\alpha+1)\left\|\frac{\delta_\mu^{\pi_\alpha^\star}}{\mu}\right\|_\infty^2.$$

In Theorem 2, we establish convergence to $\epsilon$-optimality with $\epsilon = V_\mu^\star - V_\mu^{\pi_\alpha^\star}$, which can be made arbitrarily small by taking $\alpha > 0$ to be small. However, we do not have convergence to exact optimality as $k \to \infty$, where $k$ is the iteration count. Moreover, the convergence rate is sublinear, and the constant factor depends on the sizes of the state and action spaces, which may be quite large. These shortcomings are addressed by the analysis of the natural policy gradient method presented in the next section.

## 6 Convergence of Natural Policy Gradient

In this section, we study convergence of natural policy gradient with softmax parametrization:

$$\pi_\theta(a\,|\,s) = \frac{\exp(\theta_{s,a})}{\sum_{a'}\exp(\theta_{s,a'})}.$$

where $\theta \in \mathbb{R}^{|\mathcal{S}|\times|\mathcal{A}|}$. The softmax parametrized policy automatically satisfies $\pi_\theta \in \Pi_+$, so a projection mechanism is no longer necessary.

For a given $\mu$ with full support, the natural policy gradient algorithm, which can be seen as a policy gradient with the Fisher information matrix, is

$$F_\mu(\theta^k) = \sum_{s\in\mathcal{S}}\delta_\mu^{\pi_\theta}(s)\mathbb{E}_{a\sim\pi_\theta(\cdot|s)}\left[\nabla_\theta\log\pi_\theta(a\,|\,s)\left(\nabla_\theta\log\pi_\theta(a\,|\,s)\right)^\top\right] \quad \text{for } k = 0,1,\dots,$$

$$\theta^{k+1} = \theta^k + \eta_k F_\mu(\theta^k)^\dagger\nabla_\theta V_\mu^{\pi_k}$$

where $\dagger$ denotes the Moore–Penrose pseudoinverse. It is known that natural policy gradient algorithm can also be expressed as

$$\pi_{k+1}(a\,|\,s) = \pi_k(a\,|\,s)\frac{\exp(\eta_k Q^{\pi_k}(s,a))}{z_s^k} \propto \pi_0(a\,|\,s)\exp\left(\sum_{i=0}^k \eta_i Q^{\pi_i}(s,a)\right)$$

for all $s \in \mathcal{S}$, $a \in \mathcal{A}$, and $k = 0,1,\dots$, where

$$z_s^k = \sum_{a\in\mathcal{A}}\pi_k(a\,|\,s)\exp(\eta_k Q^{\pi_k}(s,a))$$

is a normalization factor. In the online learning literature, this update rule is also known as multiplicative weights updates (Freund & Schapire, 1997) and the update rule does not depend on the initial state distribution $\mu$, as the pseudoinverse of the Fisher information removes this dependency.

## 6.1 SUBLINEAR CONVERGENCE WITH CONSTANT STEP SIZE

We establish the sublinear convergence of the policy gradient algorithm with a constant step size. As a first step in our analysis, we state the following lemma, which ensures that the policies generated by the natural policy gradient algorithm improve monotonically.

**Lemma 7.** *Under Assumption 1, for $\pi_0 \in \Pi_+$ and given $\mu$ with full support, the natural policy gradient with constant step size $\eta > 0$ generates a sequence of policies $\{\pi_k\}_{k=1}^{\infty}$ satisfying*

$$V_\mu^{\pi_k} \leq V_\mu^{\pi_{k+1}}.$$

Next, define

$$\mathrm{KL}_{\delta_\mu^\pi}(\pi, \pi') = \sum_{s \in \mathcal{S}} \delta_\mu^\pi(s) \mathrm{KL}\big(\pi(\cdot \mid s),\, \pi_k(\cdot \mid s)\big),$$

where $\mathrm{KL}(p, q) = \sum_{i=1}^n p_i \log(p_i/q_i)$ for $p, q \in \mathcal{M}(\mathcal{A})$, and also define $\left\| (I - T^\pi)^{-1} \right\|_\infty = C_\pi < \infty$. Combining Lemmas 5 and 7, we obtain the following sublinear convergence.

**Theorem 3.** *Under Assumption 1, for $\pi_0 \in \Pi_+$ and given $\mu$ with full support, the natural policy gradient with constant step size $\eta > 0$ generates a sequence of policies $\{\pi_k\}_{k=1}^{\infty}$ satisfying*

$$V_\mu^\pi - V_\mu^{\pi_k} \leq \frac{1}{k+1} \left( \frac{\mathrm{KL}_{\delta_\mu^\pi}(\pi, \pi_0)}{\eta} + C_\pi \big( \left\| V_+^\star \right\|_\infty + \left\| V^{\pi_0} \right\|_\infty \big) \right).$$

*Hence, $V_\mu^{\pi_k} \to V_{+,\mu}^\star$ as $k \to \infty$.*

Appendix C provides the proof, which closely follows Xiao (2022).

Unlike the projected policy gradient algorithm, the convergence rate of the policy gradient method is independent of the size of the state or action space. Moreover, if we assume the rewards are nonnegative, the convergence result can be strengthened from $V_\mu^{\pi_k} \to V_{+,\mu}^\star$ to $V_\mu^{\pi_k} \to V_\mu^\star$.

**Corollary 2.** *Assume the rewards are nonnegative. Under Assumption 1, for $\pi_0 \in \Pi_+$ and given $\mu$ with full support, the natural policy gradient algorithm with constant step size $\eta > 0$ generates a sequence of policies $\{\pi_k\}_{k=1}^{\infty}$ satisfying*

$$V_\mu^\star - V_\mu^{\pi_k} \leq \frac{1}{k+1} \left( \frac{\mathrm{KL}_{\delta_\mu^{\pi^\star}}(\pi^\star, \pi_0)}{\eta} + C_{\pi^\star} \left\| V^\star \right\|_\infty \right)$$

*for any optimal policy $\pi^\star$.*

## 6.2 LINEAR CONVERGENCE WITH ADAPTIVE STEP SIZE

Next, we present the *linear* convergence rate of the natural policy gradient algorithm with adaptive step size. For a given $\mu$ with full support, define $\vartheta_\mu^\pi = \left\| \frac{\delta_\mu^\pi}{\mu} \right\|_\infty \in [1, \infty)$, which represent the distribution mismatch between $\mu$ and $\delta_\mu^\pi$.

**Theorem 4.** *Under Assumption 1, for $\pi_0 \in \Pi_+$ and $\mu$ with full support, the natural policy gradient algorithm with step sizes $(\vartheta_\mu^\pi - 1)\eta_{k+1} \geq \vartheta_\mu^\pi \eta_k > 0$ generates a sequence of policies $\{\pi_k\}_{k=1}^{\infty}$ satisfying*

$$V_\mu^\pi - V_\mu^{\pi_k} \leq \left( 1 - \frac{1}{\vartheta_\mu^\pi} \right)^k \left( V_\mu^\pi - V_\mu^{\pi_0} + \frac{\mathrm{KL}_{\delta_\mu^\pi}(\pi, \pi_0)}{\eta_0(\vartheta_\mu^\pi - 1)} \right).$$

*Hence, $V_\mu^{\pi_k} \to V_{+,\mu}^\star$ as $k \to \infty$.*

As the distribution mismatch decreases, we can see that the natural policy gradient converges faster. Again, the convergence rate is independent of the size of the state or action space, and we can strengthen the convergence result if we assume the rewards are nonnegative.

**Corollary 3.** *Assume the rewards are nonnegative. Under Assumption 1, for $\pi_0 \in \Pi_+$ and given $\mu$ with full support, the natural policy gradient algorithm with step size $(\vartheta_\mu^\pi - 1)\eta_{k+1} \geq \vartheta_\mu^\pi \eta_k > 0$ generates a sequence of policies $\{\pi_k\}_{k=1}^{\infty}$ satisfying*

$$V_\mu^\star - V_\mu^{\pi_k} \leq \left( 1 - \frac{1}{\vartheta_\mu^{\pi^\star}} \right)^k \left( V_\mu^\star - V_\mu^{\pi_0} + \frac{\mathrm{KL}_{\delta_\mu^{\pi^\star}}(\pi^\star, \pi_0)}{\eta_0(\vartheta_\mu^{\pi^\star} - 1)} \right).$$

*for any optimal policy $\pi^\star$.*

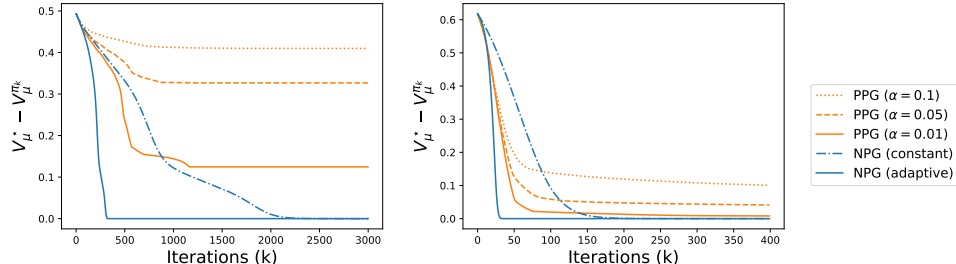

Figure 2: Comparison of projected policy gradient (PPG) and natural policy gradient (NPG) algorithms in (left) Frozenlake and (right) Cliffwalk. The limit of the projected policy gradient algorithm gets closer to the optimum as $\alpha > 0$ gets smaller.

Although the adaptive step size yields a linear convergence rate, it requires knowledge of $\vartheta^\pi_\mu$ to set the step sizes. In contrast, a constant step size always guarantees a sublinear rate.

## 7 EXPERIMENTS

For the experiments, we consider two toy examples: Frozenlake with $4 \times 4$ states and 4 actions and CliffWalk with $3 \times 7$ states and 4 actions. We use nonnegative rewards which ensures $V^\star_\mu = V^\star_{+,\mu}$ by Lemma 4, and uniform initial state distribution. Further details are provided in Appendix D.

We run the projected policy gradient method with $\alpha \in \{0.1, 0.05, 0.01\}$ and the natural policy gradient method with both constant and adaptive step sizes. All algorithms are implemented using the transient policy gradient with transient visitation measure. For Frozenlake, we use $\{0.1 \cdot 1.01^k\}^\infty_{k=0}$ for the adaptive step size of natural policy gradient, where $k$ is the number of iterations, and $0.1$ for others. For CliffWalk, we use $\{0.05 \cdot 1.1^k\}^\infty_{k=0}$ for the adaptive step size and $0.05$ for others.

The results are shown in Figure 2. The natural policy gradient with adaptive step size exhibits the fastest convergence rate among the algorithms, as the guaranteed linear rate of Corollary 3 predicts. Note that both natural policy gradients converge to $V^\star_\mu$ while the projected policy gradient converges to $V^{\pi^\star_\alpha}_\mu$ for each $\alpha$, and smaller $\alpha$ makes projected policy gradient converge closer to $V^\star_\mu$ since $V^{\pi^\star_\alpha}_\mu$ increases monotonically to $V^\star_\mu$ as $\alpha \to 0$.

Additionally, we run an experiment with pathological MDP of Figure 1, shown in Appendix D.

## 8 CONCLUSION

In this work, we present the first analysis of policy gradient methods for undiscounted expected total-reward infinite-horizon MDPs. Our approach combines the classical recurrent-transient theory from Markov chain theory with prior analysis techniques for policy gradient methods. Specifically, we first establish invariance of the classification of MDP states on $\Pi_+$, where the value function is continuous, and define a new transient visitation measure that leads to a transient policy gradient. Based on this machinery, we establish non-asymptotic convergence rates for projected policy gradient and natural policy gradient in the undiscounted total-reward setting.

Our framework opens the door to several directions for future work. One direction is to extend our results to function approximation in a sampling setting, where restricted parametric policies may not include the optimal policy, and the estimation and optimization errors from finite samples must be quantified. Another promising direction is to establish the convergence of the naive policy gradient with softmax parameterization, without preconditioning by the Fisher information matrix.

Finally, we highlight that recurrent-transient classification of MDP states is a fundamental and broadly applicable technique. Previously, the recurrent-transient theory was also applied to improve the convergence of policy iteration independently of the discount factor (Fox & Landi, 1968; Bertsekas & Tsitsiklis, 1991; Scherrer, 2013). We expect that this technique can be used to analyze a wide range of RL algorithms in the undiscounted total-reward setting.

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

## A  OMITTED PROOFS IN SECTION 4

For definitions of basic concepts of transient-recurrent theory such as irreducible class, communicating class, closedness, etc., please refer to Brémaud (2013, Chapter 2 and 3)

### A.1  PROOF OF PROPOSITION 1

*Proof.* For any $\pi, \pi' \in \Pi_+$, $P^\pi(s, s') \neq 0$ if and only if $P^{\pi'}(s, s') \neq 0$ for $s, s' \in \mathcal{S}$ by definition of $\Pi_+$. This implies $s$ and $s'$ of $P^\pi$ communicate if and only if $s$ and $s'$ of $P^{\pi'}$ communicate, and thus communicating class is invariant for $\pi \in \Pi_+$. It was known that states in communicating class are all transient or recurrent (Brémaud, 2013, Theorem 3.1.6). Next, set of states is closed in $P^\pi$ if and only if it is closed in $P^{\pi'}$. Therefore, since communicating class is closed if and only if it is recurrent in finite states MDP (Brémaud, 2013, Theorem 3.2.8), we obtain desired result.  □

### A.2  PROOF OF LEMMA 1

*Proof.* Since $\sum_{k=0}^\infty (P^\pi)^k(s, s) = \infty$ for recurrent state $s$, $r^\pi(s) = 0$ to satisfy Assumption 1.  □

By Lemma 1 and 2, we directly obtain following Corollary.

**Corollary 4.** *Under Assumption 1, $V^\pi(s) = 0$ for all recurrent states $s$.*

### A.3  PROOF OF LEMMA 3

*Proof.* $\pi \mapsto P^\pi$ is continuous, and by Proposition 1, $\pi \mapsto T^\pi$ is also continuous on $\Pi_+$. Since $(I - T^\pi)^{-1}$ is continuous with respect to $T^\pi$, by Lemma 2, $V^\pi$ and $V_\mu^\pi$ are continuous with respect to $\pi$.  □

### A.4  PROOF OF LEMMA 5

*Proof.*

$$V^\pi(s) - V^{\pi'}(s) = \sum_{a \in \mathcal{A}} Q^\pi(s, a)\pi(a \mid s) - \sum_{a \in \mathcal{A}} Q^{\pi'}(s, a)\pi'(a \mid s)$$

$$= \sum_{a \in \mathcal{A}} Q^{\pi'}(s, a)(\pi(a \mid s) - \pi'(a \mid s)) + \sum_{a \in \mathcal{A}}(Q^\pi(s, a) - Q^{\pi'}(s, a))\pi(a \mid s)$$

$$= \sum_{a \in \mathcal{A}} Q^{\pi'}(s, a)(\pi(a \mid s) - \pi'(a \mid s)) + \sum_{s' \in \mathcal{S}}\sum_{a \in \mathcal{A}} P(s' \mid s, a)(V^\pi(s') - V^{\pi'}(s'))\pi(a \mid s)$$

$$= \sum_{a \in \mathcal{A}} Q^{\pi'}(s, a)(\pi(a \mid s) - \pi'(a \mid s)) + (P^\pi(V^\pi - V^{\pi'}))(s)$$

where we used Bellman equation in third equality. Let $u(s) = \sum_{a \in \mathcal{A}} Q^{\pi'}(s, a)(\pi(a \mid s) - \pi'(a \mid s))$. Then, we have

$$V^\pi - V^{\pi'} = u + P^\pi(V^\pi - V^{\pi'})$$
$$= u + T^\pi(V^\pi - V^{\pi'})$$

which further implies

$$V^\pi - V^{\pi'} = (I - T^\pi)^{-1}u$$

and

$$V_\mu^\pi - V_\mu^{\pi'} = \mu^\intercal(I - T^\pi)^{-1}u.$$

□

**Corollary 5.** *Under Assumption 1, if $\pi, \pi' \in \Pi_+$,*

$$V_\mu^\pi - V_\mu^{\pi'} = \sum_{s' \in \mathcal{S}}\sum_{a \in \mathcal{S}} Q^{\pi'}(s', a)(\pi(a \mid s') - \pi'(a \mid s'))\delta_\mu^\pi(s').$$

*Proof.* By Lemma 1 and 2, we obtain $V^\pi(s) = 0$ for any recurrent state $s$. Thus by Proposition 1, condition of Lemma 5 is satisfied. □

**Corollary 6.** *Under Assumption 1, if rewards are nonnegative, for $\pi \in \Pi$,*

$$V_\mu^\star - V_\mu^\pi = \sum_{s' \in \mathcal{S}} \sum_{a \in \mathcal{S}} Q^\pi(s', a)(\pi^\star(a \mid s') - \pi(a \mid s'))\delta_\mu^{\pi^\star}(s').$$

*Proof.* Since $V^\star \geq V^\pi \geq 0$ by definition of $V^\pi$, $\pi$ satisfies condition of Lemma 5. □

### A.5 PROOF OF LEMMA 4

*Proof.* By Corollary 5, we have

$$V_\mu^\star - V_\mu^\pi \leq |\mathcal{S}||\mathcal{A}| \, \|Q^\pi\|_\infty \, \|\pi^\star - \pi\|_\infty \left\|\delta_\mu^{\pi^\star}\right\|_\infty.$$

Since $\|Q^\pi\|_\infty$ is bounded by $\|Q^\star\|_\infty$, $\lim_{\pi \to \pi^\star} V_\mu^\pi = V_\mu^\star$ and this implies $V_{+,\mu}^\star = V_\mu^\star$. □

### A.6 PROOF OF THEOREM 1

For the proof of Theorem 1, we first prove the following lemmas.

**Lemma 8.** *Under Assumption 1, for recurrent state $s$ of $P^\pi$ where $\pi \in \Pi_+$, $r(s, a) = 0$.*

*Proof.* If $r(s, a) \neq 0$ for some $a \in \mathcal{A}$, there exist $\pi \in \Pi_+$ such that $r^\pi(s) \neq 0$ since $\|r\|_\infty \leq R$. This is contradiction by Lemma 1. □

Now we prove Theorem 1.

*Proof.* Fix $\pi \in \Pi_+$. We first clarify differentiability of $V^\pi$. For $\triangle\pi \in \mathbb{R}^{|\mathcal{S}| \times |\mathcal{A}|}$ which represents change of policy, we define $P^{\pi + \triangle\pi}(s, s') = \sum_{a \in \mathcal{A}}(\pi + \triangle\pi)(s, a)P(s' \mid s, a)$ and $r^{\pi + \triangle\pi}(s) = \sum_{a \in \mathcal{A}}(\pi + \triangle\pi)(s, a)r(s, a)$ for all $s, s' \in \mathcal{S}$. Then, if $s$ is recurrent, $r^{\pi + \triangle\pi}(s) = 0$ by Lemma 8.

By definition of $\Pi_+$, there exist open ball $B(\pi, \epsilon)$ such that $(\pi + \triangle\pi)(a \mid s) > 0$ for all $s \in \mathcal{S}, a \in \mathcal{A}$ and $\triangle\pi \in B(\pi, \epsilon)$. Then, $P^\pi(s, s') \neq 0$ if and only if $P^{\pi + \triangle\pi}(s, s') \neq 0$ for all $s, s' \in \mathcal{S}$, and we can define perturbed transient matrix $T^{\pi + \triangle\pi}$ for $\pi \in \Pi_+$.

Since spectral radius is continuous with respect to element of matrix (Horn & Johnson, 2012, Theorem 2.4.9.2), and $T^\pi$ is continuous with respect to $\pi \in \Pi_+$, there also exist open ball $B'(\pi, \epsilon) \subset B(\pi, \epsilon)$ such that spectral radius of $T^{\pi + \triangle\pi}$ is smaller than 1. (Note that this argument is valid since set of transient class and recurrent class fixed by Proposition 1.) Thus

$$V^{\pi + \triangle\pi} = \sum_{i=0}^\infty (P^{\pi + \triangle\pi})^i r^{\pi + \triangle\pi} = \sum_{i=0}^\infty (T^{\pi + \triangle\pi})^i r^{\pi + \triangle\pi} = (I - T^{\pi + \triangle\pi})^{-1} r^{\pi + \triangle\pi},$$

and this implies well-definedness of value function on $\pi + \triangle\pi$ where $c \in \Pi_+$ and $\triangle\pi \in B'(\pi, \epsilon)$. Then, by Lemma 2, differentiability of $T^\pi$ and $r^\pi$ on $\pi \in \Pi_+$ implies differentiability of $V^\pi$, and it can be easily seen that $T^\pi$ is differentiable with respect to $\pi$ since each element of $P^\pi$ is differentiable and transient class is fixed by Proposition 1. $r^\pi$ is obviously differentiable.

Therefore,

$$\begin{aligned}
\nabla_\theta V_\mu^{\pi_\theta} &= \nabla_\theta \mu^\intercal (I - T^{\pi_\theta})^{-1} r^{\pi_\theta} \\
&= \nabla_\theta(\mu^\intercal(I - T^{\pi_\theta})^{-1})r^{\pi_\theta} + \mu^\intercal(I - T^{\pi_\theta})^{-1}\nabla_\theta r^{\pi_\theta} \\
&= \mu^\intercal(I - T^{\pi_\theta})^{-1}\frac{\partial T^{\pi_\theta}}{\partial \theta}(I - T^{\pi_\theta})^{-1}r^{\pi_\theta} + \mu^\intercal(I - T^{\pi_\theta})^{-1}\nabla_\theta r^{\pi_\theta} \\
&= \mu^\intercal(I - T^{\pi_\theta})^{-1}\frac{\partial \Theta P}{\partial \theta}V^{\pi_\theta} + \mu^\intercal(I - T^{\pi_\theta})^{-1}\frac{\partial \Theta r}{\partial \theta} \\
&= \mu^\intercal(I - T^{\pi_\theta})^{-1}\frac{\partial \Theta}{\partial \theta}(PV^{\pi_\theta} + r) \\
&= \mu^\intercal(I - T^{\pi_\theta})^{-1}\frac{\partial \Theta}{\partial \theta}Q^{\pi_\theta}
\end{aligned}$$

where third equality comes from matrix calculus $\frac{\partial A^{-1}}{\partial \theta} = A^{-1} \frac{\partial A}{\partial \theta} A^{-1}$ for $A(\theta) \in \mathbb{R}^{|\mathcal{S}| \times |\mathcal{S}|}$ and forth equality is from fact that $\Theta \in \mathbb{R}^{|\mathcal{S}| \times |\mathcal{S}||\mathcal{A}|}$ is matrix form of policy $\pi_\theta$ satisfying $\Theta P = P^{\pi_\theta}$ and $\Theta r = r^{\pi_\theta}$ and $\frac{\partial T^{\pi_\theta}}{\partial \theta} V^{\pi_\theta} = \frac{\partial P^{\pi_\theta}}{\partial \theta} V^{\pi_\theta}$ by Corollary 4 and Proposition 1. $\qquad\square$

# B OMITTED PROOFS IN SECTION 5

## B.1 PROOF OF LEMMA 6

*Proof.* We basically follow the proof strategy of Agarwal et al. (2021); Mei et al. (2020) . Let $\theta_\beta = \theta + \beta u$. Then, with direct parametrization,

$$\max_{\|u\|_2=1} \sum_a \left[ \frac{\partial \pi_{\theta_\beta}(a|s)}{\partial \beta} \Big|_{\beta=0} \right] \leq \sqrt{|\mathcal{A}|}, \qquad \sum_a \left[ \frac{\partial^2 \pi_{\theta_\beta}(a|s)}{\partial \beta^2} \Big|_{\beta=0} \right] = 0.$$

Note that $T^{\pi_{\theta_\beta}} \in \mathbb{R}^{|\mathcal{S}| \times |\mathcal{S}|}$ as

$$[T^{\pi_{\theta_\beta}}]_{(s,s')} = \sum_a \pi_{\theta_\beta}(a|s) \cdot P(s'|s,a) \qquad \text{for all } s, s' \in \mathcal{T}$$

$$= 0 \qquad\qquad\qquad\qquad \text{otherwise}$$

where $\mathcal{T}$ is invariant transient class. Then, the derivative with respect to $\beta$ is

$$\left[ \frac{\partial T^{\pi_{\theta_\beta}}}{\partial \beta} \Big|_{\beta=0} \right]_{(s,s')} = \sum_a \left[ \frac{\partial \pi_{\theta_\beta}(a|s)}{\partial \beta} \Big|_{\beta=0} \right] \cdot P(s'|s,a)$$

for $s, s' \in \mathcal{T}$, and for any vector $x \in \mathbb{R}^{|\mathcal{S}|}$, we have

$$\left\| \frac{\partial T^{\pi_{\theta_\beta}}}{\partial \beta} \Big|_{\beta=0} x \right\|_\infty = \max_{s \in \mathcal{T}} \left| \sum_{s' \in \mathcal{T}} \sum_a \left[ \frac{\partial \pi_{\theta_\beta}(a|s)}{\partial \beta} \Big|_{\beta=0} \right] \cdot P(s'|s,a) \cdot x(s') \right|$$

$$\leq \max_{s \in \mathcal{T}} \sum_a \sum_{s' \in \mathcal{T}} P(s'|s,a) \cdot \left| \frac{\partial \pi_{\theta_\beta}(a|s)}{\partial \beta} \Big|_{\beta=0} \right| \cdot \|x\|_\infty$$

$$\leq \max_{s \in \mathcal{T}} \sum_a \left| \frac{\partial \pi_{\theta_\beta}(a|s)}{\partial \beta} \Big|_{\beta=0} \right| \cdot \|x\|_\infty.$$

Therefore,

$$\max_{\|u\|_2=1} \left\| \frac{\partial T^{\pi_{\theta_\beta}}}{\partial \beta} \Big|_{\beta=0} x \right\|_\infty \leq \sqrt{\mathcal{A}} \cdot \|x\|_\infty$$

Similarly, taking second derivative with respect to $\beta$,

$$\left[ \frac{\partial^2 T^{\pi_{\theta_\beta}}}{\partial \beta^2} \Big|_{\beta=0} \right]_{(s,s')} = \sum_a \left[ \frac{\partial^2 \pi_{\theta_\beta}(a|s)}{\partial \beta^2} \Big|_{\beta=0} \right] \cdot P(s'|s,a) = 0.$$

Next, consider the state value function of $\pi_{\theta_\beta}$,

$$V^{\pi_{\theta_\beta}}(s) = e_s^\top M^{\pi_{\theta_\beta}} r^{\pi_\beta},$$

where

$$M^{\pi_{\theta_\beta}} = (I - T^{\pi_{\theta_\beta}})^{-1}, \qquad r^{\pi_{\theta_\beta}}(s) = \sum_a \pi_{\theta_\beta}(a|s) \cdot r(s,a) \quad \text{for all } s \in \mathcal{S}.$$

Since $[T^{\pi_{\theta_\beta}}]_{(s,s')} \geq 0$, for all $s, s'$, we have $[M^{\pi_{\theta_\beta}}]_{(s,s')} \geq 0$. Suppose $\|M^{\pi_{\theta_\beta}}\|_\infty \leq C_\beta$. Then, for any vector $x \in \mathbb{R}^S$,

$$\|M^{\pi_{\theta_\beta}} x\|_\infty \leq C_\beta \cdot \|x\|_\infty.$$

Note that $\|r^{\pi_{\theta_\beta}}\|_\infty \leq R$. Thus, we have

$$\left\|\frac{\partial r^{\pi_{\theta_\beta}}}{\partial \beta}\right\|_\infty = \max_s \left|\frac{\partial r^{\pi_{\theta_\beta}}(s)}{\partial \beta}\right|$$

$$\leq R \max_s \sum_a \left[\frac{\partial \pi_{\theta_\beta}(a|s)}{\partial \beta}\Big|_{\beta=0}\right].$$

Then

$$\max_{\|u\|_2=1} \left\|\frac{\partial r^{\pi_{\theta_\beta}}}{\partial \beta}\right\|_\infty \leq R\sqrt{\mathcal{A}}.$$

Similarly,

$$\left\|\frac{\partial^2 r^{\pi_{\theta_\beta}}}{\partial \beta^2}\right\|_\infty \leq \max_s R \sum_a \left[\frac{\partial^2 \pi_{\theta_\beta}(a|s)}{\partial \beta^2}\Big|_{\beta=0}\right]$$

$$= 0.$$

Derivative of value state function with respect to $\beta$ is

$$\frac{\partial V^{\pi_{\theta_\beta}}(s)}{\partial \beta} = e_s^\top M^{\pi_{\theta_\beta}} \frac{\partial T^{\pi_{\theta_\beta}}}{\partial \beta} M^{\pi_{\theta_\beta}} r^{\pi_{\theta_\beta}} + e_s^\top M^{\pi_{\theta_\beta}} \frac{\partial r^{\pi_{\theta_\beta}}}{\partial \beta}.$$

by matrix calculus $\frac{\partial A^{-1}}{\partial \theta} = A^{-1} \frac{\partial A}{\partial \theta} A^{-1}$. Taking second derivative w.r.t. $\beta$,

$$\frac{\partial^2 V^{\pi_{\theta_\beta}}(s)}{\partial \beta^2} = 2 \cdot e_s^\top M^{\pi_{\theta_\beta}} \frac{\partial T^{\pi_{\theta_\beta}}}{\partial \beta} M^{\pi_{\theta_\beta}} \frac{\partial T^{\pi_{\theta_\beta}}}{\partial \beta} M^{\pi_{\theta_\beta}} r^{\pi_{\theta_\beta}} + e_s^\top M^{\pi_{\theta_\beta}} \frac{\partial^2 T^{\pi_{\theta_\beta}}}{\partial \beta^2} M^{\pi_{\theta_\beta}} r^{\pi_{\theta_\beta}}$$

$$+ 2 \cdot e_s^\top M^{\pi_{\theta_\beta}} \frac{\partial T^{\pi_{\theta_\beta}}}{\partial \beta} M^{\pi_{\theta_\beta}} \frac{\partial r^{\pi_{\theta_\beta}}}{\partial \beta} + e_s^\top M^{\pi_{\theta_\beta}} \frac{\partial^2 r^{\pi_{\theta_\beta}}}{\partial \beta^2}.$$

For the last term,

$$\left|e_s^\top M^{\pi_{\theta_\beta}} \frac{\partial^2 r^{\pi_{\theta_\beta}}}{\partial \beta^2}\Big|_{\beta=0}\right| = 0.$$

For the second last term,

$$\max_{\|u\|_2=1} \left|e_s^\top M^{\pi_{\theta_\beta}} \frac{\partial T^{\pi_{\theta_\beta}}}{\partial \beta} M^{\pi_{\theta_\beta}} \frac{\partial r^{\pi_{\theta_\beta}}}{\partial \beta}\Big|_{\beta=0}\right| \leq \max_{\|u\|_2=1} \left\|M^{\pi_{\theta_\beta}} \frac{\partial T^{\pi_{\theta_\beta}}}{\partial \beta} M^{\pi_{\theta_\beta}} \frac{\partial r^{\pi_{\theta_\beta}}}{\partial \beta}\Big|_{\beta=0}\right\|_\infty$$

$$\leq C_\beta \max_{\|u\|_2=1} \left\|\frac{\partial T^{\pi_{\theta_\beta}}}{\partial \beta} M^{\pi_{\theta_\beta}} \frac{\partial r^{\pi_{\theta_\beta}}}{\partial \beta}\Big|_{\beta=0}\right\|_\infty$$

$$\leq C_\beta \sqrt{|\mathcal{A}|} \max_{\|u\|_2=1} \|M^{\pi_{\theta_\beta}} \frac{\partial r^{\pi_{\theta_\beta}}}{\partial \beta}\Big|_{\beta=0}\|_\infty$$

$$\leq C_\beta^2 \sqrt{|\mathcal{A}|} \max_{\|u\|_2=1} \left\|\frac{\partial r^{\pi_{\theta_\beta}}}{\partial \beta}\Big|_{\beta=0}\right\|_\infty$$

$$\leq C_\beta^2 |\mathcal{A}| R.$$

For the first term, similarly,

$$\max_{\|u\|_2=1} \left|e_s^\top M^{\pi_{\theta_\beta}} \frac{\partial T^{\pi_{\theta_\beta}}}{\partial \beta} M^{\pi_{\theta_\beta}} \frac{\partial T^{\pi_{\theta_\beta}}}{\partial \beta} M^{\pi_{\theta_\beta}} r^{\pi_{\theta_\beta}}\Big|_{\beta=0}\right| \leq C_\beta \cdot \sqrt{\mathcal{A}} \cdot C_\beta \sqrt{\mathcal{A}} \cdot C_\beta \cdot R$$

$$= C_\beta^3 R |\mathcal{A}|.$$

For the second term,

$$\left|e_s^\top M^{\pi_{\theta_\beta}} \frac{\partial^2 P^{\pi_{\theta_\beta}}}{\partial \beta^2} M^{\pi_{\theta_\beta}} r^{\pi_{\theta_\beta}}\Big|_{\beta=0}\right| = 0$$

Finally, we have

$$
\left| \frac{\partial^2 V^{\pi_{\theta_\beta}}(s)}{\partial \beta^2} \right|_{\beta=0} \le 2 \cdot \left| e_s^\top M^{\pi_{\theta_\beta}} \frac{\partial T^{\pi_{\theta_\beta}}}{\partial \beta} M^{\pi_{\theta_\beta}} \frac{\partial T^{\pi_{\theta_\beta}}}{\partial \beta} M^{\pi_{\theta_\beta}} r^{\pi_{\theta_\beta}} \right|_{\beta=0}
$$

$$
+ \left| e_s^\top M^{\pi_{\theta_\beta}} \frac{\partial^2 T^{\pi_{\theta_\beta}}}{\partial \beta^2} M^{\pi_{\theta_\beta}} r^{\pi_{\theta_\beta}} \right|_{\beta=0}
$$

$$
+ 2 \left| e_s^\top M^{\pi_{\theta_\beta}} \frac{\partial T^{\pi_{\theta_\beta}}}{\partial \beta} M^{\pi_{\theta_\beta}} \frac{\partial r^{\pi_{\theta_\beta}}}{\partial \beta} \right|_{\beta=0} + \left| e_s^\top M^{\pi_{\theta_\beta}} \frac{\partial^2 r^{\pi_{\theta_\beta}}}{\partial \beta^2} \right|_{\beta=0}
$$

$$
\le 2 C_\beta^2 (C_\beta + 1) R |\mathcal{A}|.
$$

$\square$

We now prove another key lemma for Theorem 2.

**Lemma 9** (Gradient domination). *Under Assumption 1, for $\pi \in \Pi_\alpha$,*

$$
V_\mu^{\pi_\alpha^\star} - V_\mu^\pi \le \left\| \frac{\delta_\mu^{\pi^\star}}{\mu} \right\|_\infty \max_{\bar\pi \in \Pi_\alpha} (\bar\pi - \pi)^\top \nabla_\pi V_\mu^\pi.
$$

*Proof.* We basically follow the proof strategy of Agarwal et al. (2021). Let $A^\pi(s,a) = Q^\pi(s,a) - V^\pi(s)$. Then, we have

$$
V_\mu^{\pi_\alpha^\star} - V_\mu^\pi = \sum_{s,a} \delta_\mu^{\pi^\star}(s) \pi^\star(a|s) A^\pi(s,a)
$$

$$
\le \sum_s \delta_\mu^{\pi^\star}(s) \max_{\bar\pi \in \Pi_\alpha} \left( \sum_a \bar\pi(a \mid s) A^\pi(s,a) \right)
$$

$$
\le \left( \max_s \frac{\delta_\mu^{\pi^\star}(s)}{\delta_\mu^\pi(s)} \right) \sum_s \delta_\mu^\pi(s) \max_{\bar\pi \in \Pi_\alpha} \left( \sum_a \bar\pi(a \mid s) A^\pi(s,a) \right),
$$

(1)

where first equality comes from Corollary 5, the last inequality follows since $\max_{\bar\pi \in \Pi_\alpha} \left( \sum_a \bar\pi(a \mid s) A^\pi(s,a) \right) \ge 0$ for all $s \in \mathcal{S}$ and policies $\pi \in \Pi_\alpha$. Also, we have

$$
\sum_s \delta_\mu^\pi(s) \max_{\bar\pi \in \Pi_\alpha} \left( \sum_a \bar\pi(a \mid s) A^\pi(s,a) \right) = \max_{\bar\pi \in \Pi_\alpha} \sum_{s,a} \delta_\mu^\pi(s) \bar\pi(a|s) A^\pi(s,a)
$$

$$
= \max_{\bar\pi \in \Pi_\alpha} \sum_{s,a} \delta_\mu^\pi(s) \big(\bar\pi(a|s) - \pi(a|s)\big) A^\pi(s,a)
$$

$$
= \max_{\bar\pi \in \Pi_\alpha} \sum_{s,a} \delta_\mu^\pi(s) \big(\bar\pi(a|s) - \pi(a|s)\big) Q^\pi(s,a)
$$

$$
= \max_{\bar\pi \in \Pi_\alpha} (\bar\pi - \pi)^\top \nabla_\pi V_\mu^\pi.
$$

where the first equality follows from the fact that $\max_{\bar\pi}$ is attained at an action which maximizes $A^\pi(s,\cdot)$, the second equality is from $\sum_a \pi(a|s) A^\pi(s,a) = 0$, the third equality is from $\sum_a (\bar\pi(a|s) - \pi(a|s)) V^\pi(s) = 0$ for all $s$, and the last equality follows from the Theorem 1 with direct parameterization. Finally,

$$
V_\mu^{\pi_\alpha^\star} - V_\mu^\pi \le \left\| \frac{\delta_\mu^{\pi^\star}}{\delta_\mu^\pi} \right\|_\infty \max_{\bar\pi \in \Pi_\alpha} (\bar\pi - \pi)^\top \nabla_\pi V_\mu^\pi
$$

$$
\le \left\| \frac{\delta_\mu^{\pi^\star}}{\mu} \right\|_\infty \max_{\bar\pi \in \Pi_\alpha} (\bar\pi - \pi)^\top \nabla_\pi V_\mu^\pi.
$$

where the last step follows from $\max_{\bar\pi \in \Pi_\alpha} (\bar\pi - \pi)^\top \nabla_\pi V_\mu^\pi \ge 0$ for any policy $\pi$ and $\delta_\mu^\pi(s) \ge \mu(s)$. $\square$

### B.2 Proof of Theorem 2

Following the proof strategy of Xiao (2022), we consider composite optimization problem:

$$\min_{x \in \mathbb{R}^n} F(x) := f(x) + \Psi(x)$$

where $f$ is $L$-smooth and $\Psi$ is is proper, convex, and closed (Rockafellar, 1970). Define $F^\star = \min_x F(x)$.

For convex function $\phi$, define proximal operator as

$$\mathbf{prox}_\phi(x) = \operatorname{argmin}_y \{\phi(y) + \frac{1}{2}\|y - x\|_2^2\}.$$

Then, for composite optimization problem, proximal gradient method is

$$x^{k+1} = \mathbf{prox}_{\eta_k \Psi}(x^k - \eta_k \nabla f(x^k)).$$

Specifically, let $\eta_k = \frac{1}{L}$. and define

$$T_L(x) = \mathbf{prox}_{\frac{1}{L}\Psi}(x - \frac{1}{L}\nabla f(x))$$

such that proximal gradient method can be expressed as $x^{k+1} = T_L(x^k)$. We also define gradient mapping

$$G_L = L(x - T_L(x)).$$

**Definition** (weak gradient-mapping domination). *Consider composite optimization problem. We say that $F$ satisfies a weak gradient-mapping dominance condition, of exponent $\alpha \in (1/2, 1]$, if there exists $\omega > 0$ such that*

$$\|G_L(x)\|_2 \geq \sqrt{2\omega} \left(F(T_L(x)) - F^\star\right)^\alpha, \qquad \forall x \in \operatorname{dom}\Psi.$$

**Fact 2.** *(Xiao, 2022, Theorem 4) Consider the composite optimization problem. Suppose $F$ is weakly gradient-mapping dominant with exponent $\alpha = 1$ and parameter $\omega$. Then, for all $k \geq 0$, the proximal gradient method with step size $\eta_k = 1/L$ generates a sequence $\{x^k\}$ that satisfies*

$$F(x^k) - F^\star \leq \max\left\{\frac{8L}{\omega k}, \left(\frac{\sqrt{2}}{2}\right)^k \left(F(x^0) - F^\star\right)\right\}.$$

**Fact 3.** *(Nesterov, 2013, Theorem 1) Consider the composite optimization problem where $f$ is $L$-smooth on closed convex set $C$ and $\Psi$ is indicator function with $C$. Then, for $x, y \in C$,*

$$\langle \partial F(T_L(y)), T_L(y) - x \rangle \leq 2\|G_L(y)\|_2 \|T_L(y) - x\|_2.$$

Note that if $\Psi$ is indicator function with convex closed non empty subset of $C \in \mathbb{R}^n$, $\Psi$ is closed, convex, and proper and $\mathbf{prox}_{\eta\Psi} = \mathbf{proj}_C$ where $\mathbf{proj}$ is projection operator.

Now, we apply previous results to our projected policy gradient setup. Let $-\Psi$ be indicator function with $\Pi_\alpha$ and $f(\pi) = -V_\mu^\pi$. Then $\pi^{k+1} = T_L(\pi^k)$ is projected policy gradient.

To prove Theorem 2, we first prove following lemma.

**Lemma 10.** *Under Assumption 1, for a given $\mu$ with full support, suppose that $V_\mu^\pi$ is $L$-smooth for $\pi \in \Pi_\alpha$. Then,*

$$V_\mu^{\pi_\alpha^\star} - V_\mu^{T_L(\pi)} \leq 2\sqrt{2|\mathcal{S}|} \left\|\frac{\delta_\mu^{\pi_\alpha^\star}}{\mu}\right\|_\infty \|G_L(\pi)\|_2.$$

*Proof.* By Fact 3, for all $\pi, \pi' \in \Pi_\alpha$,

$$\langle \nabla V_\mu^{T_L(\pi)}, \pi' - T_L(\pi) \rangle \leq 2\|G_L(\pi)\|_2 \|T_L(\pi) - \pi'\|_2.$$

Since $T_L(\pi) \in \Pi_\alpha$ and $\|\pi_1 - \pi_2\|_2 \leq \sqrt{2|\mathcal{S}|}$ for any $\pi_1, \pi_2 \in \Pi_\alpha$, we obtain

$$\max_{\pi' \in \Pi_\alpha} \langle \nabla V_\mu^{T_L(\pi)}, \pi' - T_L(\pi) \rangle \leq 2\sqrt{2|\mathcal{S}|} \|G_L(\pi)\|_2.$$

Combining the above bound with Lemma 9 gives the desired inequality. $\qquad\square$

Now we are ready to Theorem 2.

*Proof.* By lemma 10, weak gradient-mapping domination condition holds with

$$\alpha = 1, \qquad \omega = \frac{1}{16\,|\mathcal{S}|} \left\| \frac{\delta_\mu^{\pi_\alpha^\star}}{\mu} \right\|_\infty^{-2}, \qquad L = 2RC_\alpha^2(C_\alpha + 1)|\mathcal{A}|.$$

Then, by applying Fact 2, we get

$$V_\mu^{\pi_\alpha^\star} - V_\mu^{\pi_k} \le \frac{8L}{\omega k} = \frac{256R|\mathcal{S}|\,|\mathcal{A}|C_\alpha^2(C_\alpha + 1)}{k} \left\| \frac{\delta_\mu^{\pi_\alpha^\star}}{\mu} \right\|_\infty^2.$$

Lastly, in Fact 2, exponential decay part is always smaller than the sublinear part. □

### B.3 Proof of Corollary 1

*Proof.* Let $\alpha = \frac{\epsilon}{2|\mathcal{S}||\mathcal{A}|\left\|\delta_\mu^{\pi^\star}\right\|_\infty \|Q^{\pi^\star}\|_\infty}$. Since there exist $\pi \in \Pi_\alpha$ such that $\|\pi^\star - \pi\|_\infty \le \alpha$, by Corollary 6, we have

$$V_\mu^\star - V_\mu^\pi = \sum_{s' \in \mathcal{S}} \sum_{a \in \mathcal{S}} Q^\pi(s', a)(\pi^\star(a \mid s') - \pi(a \mid s'))\delta_\mu^{\pi^\star}(s')$$

$$\le |\mathcal{S}||\mathcal{A}| \left\|\delta_\mu^{\pi^\star}\right\|_\infty \|Q^\pi\|_\infty \alpha$$

$$\le \frac{\epsilon}{2}$$

and this implies $V_\mu^\star - V_\mu^{\pi_\alpha^\star} \le \frac{\epsilon}{2}$. and the result comes from Theorem 2 by having $V_\mu^{\pi_\alpha^\star} - V_\mu^{\pi_k} \le \frac{\epsilon}{2}$. □

## C Omitted proofs in Section 6

### C.1 Reformulation of natural policy gradient

We basically follow the derivation in Agarwal et al. (2021). First, we provide explicit form of policy gradient with softmax parametrization.

**Lemma 11.**
$$\frac{\partial V_\mu^{\pi_\theta}}{\partial \theta_{s,a}} = \delta_\mu^\pi(s)\pi_\theta(a \mid s)(Q^{\pi_\theta}(s, a) - V^{\pi_\theta}(s)).$$

*Proof.* With the softmax policy parameterization, we have

$$\frac{\partial \log \pi_\theta(a \mid s)}{\partial \theta_{s',a'}} = \mathbf{1}[s = s']\left(\mathbf{1}[a = a'] - \pi_\theta(a' \mid s)\right)$$

where $\mathbf{1}$ is indicator function. Then, by Theorem 1,

$$\frac{\partial V_\mu^{\pi_\theta}}{\partial \theta_{s',a'}} = \sum_{s \in \mathcal{S}} \delta_\mu^\pi(s')\mathbb{E}_{a \sim \pi_\theta(\cdot|s)}\left[ Q^{\pi_\theta}(s, a) \frac{\partial \log \pi_\theta(a \mid s)}{\partial \theta_{s',a'}} \right]$$

$$= \sum_{s \in \mathcal{S}} \delta_\mu^\pi(s')\mathbb{E}_{a \sim \pi_\theta(\cdot|s)}\left[ Q^{\pi_\theta}(s, a)\,\mathbf{1}[s = s']\left(\mathbf{1}[a = a'] - \pi_\theta(a' \mid s)\right) \right]$$

$$= \delta_\mu^\pi(s')\mathbb{E}_{a \sim \pi_\theta(\cdot|s')}\left[ Q^{\pi_\theta}(s', a)\left(\mathbf{1}[a = a'] - \pi_\theta(a' \mid s')\right) \right]$$

$$= \delta_\mu^\pi(s')\left( \mathbb{E}_{a \sim \pi_\theta(\cdot|s')}\left[ Q^{\pi_\theta}(s', a)\mathbf{1}[a = a'] \right] - \pi_\theta(a' \mid s')\mathbb{E}_{a \sim \pi_\theta(\cdot|s')}\left[ Q^{\pi_\theta}(s', a) \right] \right)$$

$$= \delta_\mu^\pi(s')\,\pi_\theta(a' \mid s')(Q^{\pi_\theta}(s', a') - V^{\pi_\theta}(s')).$$

□

Now, we derive the reformulation of natural policy gradient.

*Proof.* First, we have

$$w^\top \nabla_\theta \log \pi_\theta(a \mid s) = w_{s,a} - \sum_{a' \in \mathcal{A}} w_{s,a'} \pi_\theta(a' \mid s).$$

Let $\overline{w}_s = \sum_{a' \in \mathcal{A}} w_{s,a'} \pi_\theta(a' \mid s)$, which is independent of $a$. Then,

$$\mathcal{F}_\mu(\theta)w = \sum_{s \in \mathcal{S}} \delta_\mu^{\pi_\theta}(s) \mathbb{E}_{a \sim \pi_\theta(\cdot \mid s)} \big[ \nabla_\theta \log \pi_\theta(a \mid s) \big( w^\top \nabla_\theta \log \pi_\theta(a \mid s) \big) \big]$$

$$= \sum_{s \in \mathcal{S}} \delta_\mu^{\pi_\theta}(s) \mathbb{E}_{a \sim \pi_\theta(\cdot \mid s)} \big[ \nabla_\theta \log \pi_\theta(a \mid s) \big( w_{s,a} - \overline{w}_s \big) \big]$$

$$= \sum_{s \in \mathcal{S}} \delta_\mu^{\pi_\theta}(s) \mathbb{E}_{a \sim \pi_\theta(\cdot \mid s)} \big[ w_{s,a} \nabla_\theta \log \pi_\theta(a \mid s) \big],$$

where the last equality uses log derivative trick with $\overline{w}_s$, and this implies that

$$\big[ \mathcal{F}_\mu(\theta)w \big]_{s',a'} = \delta_\mu^{\pi_\theta}(s') \, \pi_\theta(a' \mid s') \big( w_{s',a'} - \overline{w}_{s'} \big).$$

By property of Moore–Penrose pseudoinverse, $\big( \mathcal{F}_\mu(\theta) \big)^\dagger \nabla V_\mu^{\pi_\theta}$ is the minimum–norm solution of $\min_w \big\| \nabla V_\mu^{\pi_\theta} - \mathcal{F}_\mu(\theta)w \big\|_2^2$ where $\| \cdot \|_2$ is Euclidean norm. By lemma 11, we have

$$\big\| \nabla V_\mu^{\pi_\theta} - \mathcal{F}_\mu(\theta)w \big\|^2 = \sum_{s,a} \left( \delta_\mu^{\pi_\theta}(s) \pi_\theta(a \mid s) \left( Q^{\pi_\theta}(s,a) - V^{\pi_\theta}(s) - w_{s,a} - \sum_{a' \in \mathcal{A}} w_{s,a'} \pi_\theta(a' \mid s) \right) \right)^2.$$

If $w = Q^{\pi_\theta}(s,a) - V^{\pi_\theta}(s)$, $\big\| \nabla V_\mu^{\pi_\theta} - \mathcal{F}_\mu(\theta)w \big\|^2 = 0$, and $w$ has the form of $Q^{\pi_\theta}(s,a) + v(s)$ where $v$ only depends on state. Therefore, plugging $\mathcal{F}_\mu(\theta)^\dagger \nabla V_\mu^{\pi_\theta} = Q^{\pi_\theta}(s,a) + v(s)$ into the original form of natural policy gradient, we obtain reformulation of natural policy gradient.

$\square$

Following the proof strategy of Xiao (2022), we consider mirror descent framework.

Let $h : \mathcal{M}(\mathcal{A}) \to \mathbb{R}$ be a strictly convex function and continuously differentiable on the (relative) interior of $\mathcal{M}(\mathcal{A})$, denoted as $\mathrm{rint}\, \mathcal{M}(\mathcal{A})$. Define Bregman divergence generated by $h$ as

$$D(p, p') = h(p) - h(p') - \langle \nabla h(p'), \, p - p' \rangle,$$

for any $p \in \mathcal{M}(\mathcal{A})$ and $p' \in \mathrm{rint}\, \mathcal{M}(\mathcal{A})$. Specifically, Kullback–Leibler (KL) divergence, generated by the negative entropy $h(p) = \sum_{a \in \mathcal{A}} p_a \log p_a$ formulated as $D(p, p') = \sum_{a \in \mathcal{A}} p_a \log \frac{p_a}{p'_a}$.

For any $\mu \in \mathcal{M}(\mathcal{S})$, we define a weighted divergence function

$$D_\mu(\pi, \pi') = \sum_{s \in \mathcal{S}} \mu(s) \, D(\pi(\cdot \mid s), \pi'(\cdot \mid s)).$$

Following the derivations of Shani et al. (2020); Xiao (2022), we consider policy mirror descent methods with dynamically weighted divergences:

$$\pi_{k+1} = \arg\min_{\pi \in \Pi} \big\{ -\eta_k \big\langle \nabla V_\mu^{\pi_k}, \, \pi \big\rangle + D_{\delta_\mu(\pi_k)} \big( \pi, \pi_k \big) \big\},$$

where $\eta_k$ is the step size, $\mu \in \mathcal{M}_+(\mathcal{S})$ is an arbitrary state distribution.

Consider direction parametrization, and by Theorem 1, we have

$$\pi_{k+1} = \arg\min_{\pi \in \Pi} \left\{ -\eta_k \sum_{s \in \mathcal{S}} \delta_\mu^{\pi_k}(s) \big( \sum_{s \in \mathcal{S}} Q^{\pi_k}(s,a)\pi(a \mid s) + D\big( \pi(\cdot \mid s), \pi_k(\cdot \mid s) \big) \right\}$$

$$= \arg\min_{\pi \in \Pi} \left\{ -\eta_k \sum_{s \in \mathcal{S}} \big( \sum_{s \in \mathcal{S}} Q^{\pi_k}(s,a)\pi(a \mid s) + D\big( \pi(\cdot \mid s), \pi_k(\cdot \mid s) \big) \right\}$$

and this is reduced to

$$\pi_{k+1}(\cdot \,|\, s) = \arg\min_{p \in \mathcal{M}(\mathcal{A})} \Big\{ -\eta_k \sum_{a \in \mathcal{A}} Q^{\pi_k}(a, s) p(a) + D\big(p, \, \pi_k(\cdot \,|\, s)\big) \Big\}$$

for all $s \in \mathcal{S}$. It was known that solution form of this update is natural policy gradient with softmax parameterization (Beck, 2017, Section 9.1).

We say function $h$ is of Legendre type if it is essentially smooth and strictly convex in the (relative) interior of $\operatorname{dom} h$ and $h$ is essential smoothness if $h$ is differentiable and $\|\nabla h(x_k)\| \to \infty$ for every sequence $\{x_k\}$ converging to a boundary point of $\operatorname{dom} h$.

**Fact 4** (Three-point descent lemma). *(Xiao, 2022, Lemma 6) Suppose that $\mathcal{C} \subset \mathbb{R}^n$ is a closed convex set, $\phi : \mathcal{C} \to \mathbb{R}$ is a proper, closed, and convex function, $D(\cdot, \cdot)$ is the Bregman divergence generated by a function $h$ of Legendre type and $\operatorname{rint} \operatorname{dom} h \cap \mathcal{C} \neq \varnothing$. For any $x \in \operatorname{rint} \operatorname{dom} h$, let*

$$x^+ = \arg\min_{u \in \mathcal{C}} \big\{ \phi(u) + D(u, x) \big\}.$$

*Then, $x^+ \in \operatorname{rint} \operatorname{dom} h \cap \mathcal{C}$ and for any $u \in \mathcal{C}$,*

$$\phi(x^+) + D(x^+, x) \;\leq\; \phi(u) + D(u, x) - D(u, x^+).$$

In our setup, $\mathcal{C} = \mathcal{M}(\mathcal{A})$, $\phi(p) = -\eta_k \langle Q^{\pi_k}(\cdot, s), p(\cdot) \rangle$, and $h$ is the negative-entropy function, which is also of Legendre type, satisfying $\operatorname{rint} \operatorname{dom} h \cap \mathcal{C} = \operatorname{rint} \mathcal{M}(\mathcal{A}) = \operatorname{rint} \operatorname{dom} h$. Therefore, if we start with an initial point in $\operatorname{rint} \mathcal{M}(\mathcal{A})$, then every iterate will stay in $\operatorname{rint} \mathcal{M}(\mathcal{A})$.

## C.2 PROOF OF LEMMA 7

We proved more detailed version of Lemma 7.

**Lemma 12.** *Under Assumption 1, for arbitrary $\mu$ with full support, the natural policy gradient with step size $\eta_k > 0$ generates a sequence of policies $\{\pi_k\}_{k=1}^\infty$ satisfying*

$$\sum_{a \in \mathcal{A}} Q^{\pi_k}(a, s)(\pi_k(a \,|\, s) - \pi_{k+1}(a \,|\, s)) \leq 0, \qquad \forall s \in \mathcal{S},$$

*and*

$$V_\mu^{\pi_k} \leq V_\mu^{\pi_{k+1}}.$$

*Proof of Lemma 7.* Applying Fact 4 with $\mathcal{C} = \mathcal{M}(\mathcal{A})$, $\phi(p) = -\eta_k \sum_{a \in \mathcal{A}} Q^{\pi_k}(a, s) p(a)$, and KL divergence as Bregman divergence, we obtain

$$\eta_k \sum_{a \in \mathcal{A}} Q^{\pi_k}(a, s) p(a) + \mathrm{KL}\big(\pi_{k+1}(\cdot \,|\, s), \, \pi_k(\cdot \,|\, s)\big)$$

$$\leq \eta_k \sum_{a \in \mathcal{A}} Q^{\pi_k}(a, s) \pi_{k+1}(a \,|\, s) + \mathrm{KL}\big(p, \pi_k(\cdot \,|\, s)\big) - \mathrm{KL}\big(p, \pi_{k+1}(\cdot \,|\, s)\big)$$

for any $p \in \mathcal{M}(\mathcal{A})$. Rearranging terms and dividing both sides by $\eta_k$, we get

$$\sum_{a \in \mathcal{A}} Q^{\pi_k}(a, s)(p(a) - \pi_{k+1}(a \,|\, s)) + \frac{1}{\eta_k} \mathrm{KL}\big(\pi_{k+1}(\cdot \,|\, s), \pi_k(\cdot \,|\, s)\big)$$

$$\leq \frac{1}{\eta_k} \mathrm{KL}\big(p, \pi_k(\cdot \,|\, s)\big) - \frac{1}{\eta_k} \mathrm{KL}\big(p, \pi_{k+1}(\cdot \,|\, s)\big). \tag{$*$}$$

Letting $p = \pi_k(\cdot \,|\, s)$ in previous inequality yields

$$\sum_{a \in \mathcal{A}} Q^{\pi_k}(a, s)(\pi_k(a \,|\, s) - \pi_{k+1}(a \,|\, s)) \leq -\frac{1}{\eta_k} \mathrm{KL}\big(\pi_{k+1}(\cdot \,|\, s), \pi_k(\cdot \,|\, s)\big) - \frac{1}{\eta_k} \mathrm{KL}\big(\pi_k(\cdot \,|\, s), \pi_{k+1}(\cdot \,|\, s)\big).$$

Then, first results comes from nonnegativity of Bregman divergence and second result comes from Corollary 5. $\qquad\square$

## C.3 PROOF OF THEOREM 3

*Proof.* Consider previous inequality $(*)$. Let $p = \pi(\cdot \mid s) \in \Pi_+$ and add–subtract $\pi_k(\cdot \mid s)$ inside the inner product. Then we have

$$\sum_{a \in \mathcal{A}} Q^{\pi_k}(a, s)(\pi_k(a \mid s) - \pi_{k+1}(a \mid s)) + \sum_{a \in \mathcal{A}} Q^{\pi_k}(a, s)(\pi(a \mid s) - \pi_k(a \mid s))$$

$$\leq \frac{1}{\eta_k}\mathrm{KL}\big(\pi(\cdot \mid s), \pi_k(\cdot \mid s)\big) - \frac{1}{\eta_k}\mathrm{KL}\big(\pi(\cdot \mid s), \pi_{k+1}(\cdot \mid s)\big).$$

This implies

$$\sum_{s \in \mathcal{S}} \delta_\mu^\pi(s) \sum_{a \in \mathcal{A}} Q^{\pi_k}(a, s)(\pi_k(a \mid s) - \pi_{k+1}(a \mid s)) + \sum_{s \in \mathcal{S}} \delta_\mu^\pi(s) \sum_{a \in \mathcal{A}} Q^{\pi_k}(a, s)(\pi(a \mid s) - \pi_k(a \mid s))$$

$$\leq \frac{1}{\eta_k}\mathrm{KL}_{\delta_\mu^\pi}(\pi, \pi_k) - \frac{1}{\eta_k}\mathrm{KL}_{\delta_\mu^\pi}(\pi, \pi_{k+1}).$$

For the first term,

$$\sum_{s \in \mathcal{S}} \frac{\delta_\mu^\pi(s)}{\|\delta_\mu^\pi\|_\infty} \sum_{a \in \mathcal{A}} Q^{\pi_k}(a, s)(\pi_k(a \mid s) - \pi_{k+1}(a \mid s))$$

$$\geq \sum_{s \in \mathcal{S}} (\delta_\mu^\pi)'(s) \sum_{a \in \mathcal{A}} Q^{\pi_k}(a, s)(\pi_k(a \mid s) - \pi_{k+1}(a \mid s))$$

$$\geq \sum_{s \in \mathcal{S}} \delta_{(\delta_\mu^\pi)'}^{\pi_{k+1}}(s) \sum_{a \in \mathcal{A}} Q^{\pi_k}(a, s)(\pi_k(a \mid s) - \pi_{k+1}(a \mid s))$$

$$= V_{(\delta_\mu^\pi)'}^{\pi_k} - V_{(\delta_\mu^\pi)'}^{\pi_{k+1}},$$

where $(\delta_\mu^\pi)'$ is probability distribution satisfying $(\delta_\mu^\pi)'(s) \geq \frac{\delta_\mu^\pi(s)}{\|\delta_\mu^\pi\|_\infty}$, the first and second inequalities come from Lemma 12, and the last equality comes from Corollary 5.

For the second term, by Corollary 5,

$$\sum_{s \in \mathcal{S}} \delta_\mu^\pi(s) \sum_{a \in \mathcal{A}} Q^{\pi_k}(a, s)(\pi(a \mid s) - \pi_k(a \mid s)) = V_\mu^\pi - V_\mu^{\pi_k}.$$

Thus we have

$$V_\mu^\pi - V_\mu^{\pi_k} \leq \frac{1}{\eta_k}\mathrm{KL}_{\delta_\mu^\pi}(\pi, \pi_k) - \frac{1}{\eta_k}\mathrm{KL}_{\delta_\mu^\pi}(\pi, \pi_{k+1}) + \|\delta_\mu^\pi\|_\infty (V_{(\delta_\mu^\pi)'}^{\pi_{k+1}} - V_{(\delta_\mu^\pi)'}^{\pi_k}).$$

Setting $\eta_k = \eta$ for all $k \geq 0$ and summing over $k$ gives

$$\sum_{i=0}^{k} (V_\mu^\pi - V_\mu^{\pi_i}) \leq \frac{1}{\eta}\mathrm{KL}_{\delta_\mu^\pi}(\pi, \pi_0) - \frac{1}{\eta}\mathrm{KL}_{\delta_\mu^\pi}(\pi, \pi_{k+1}) + \|\delta_\mu^\pi\|_\infty (V_{(\delta_\mu^\pi)'}^{\pi_{k+1}} - V_{(\delta_\mu^\pi)'}^{\pi_0}).$$

Since $V_\mu^{\pi_k}$ are non-decreasing by Lemma 7 and KL-divergence is non-negative, we conclude that

$$V_\mu^\pi - V_\mu^{\pi_k} \leq \frac{1}{k+1}\left(\frac{\mathrm{KL}_{\delta_\mu^\pi}(\pi, \pi_0)}{\eta} + \|\delta_\mu^\pi\|_\infty (\|V_+^\star\|_\infty + \|V^{\pi_0}\|_\infty)\right).$$

For given $\eta, \epsilon > 0$, there exist $\pi$ such that $V_{+,\mu}^\star - V^\pi < \epsilon/2$ since $V_{+,\mu}^\star < \infty$. Then, by previous inequality, there exist $\pi_k$ such that $V^\pi - V_\mu^{\pi_k} < \epsilon/2$. Thus, we have $V_{+,\mu}^\star - V_\mu^{\pi_k} < \epsilon$. Since this holds for arbitrary $\epsilon$, we get $V_\mu^{\pi_k} \to V_{+,\mu}^\star$.

$\square$

## C.4 PROOF OF COROLLARY 2

*Proof.* In the previous proof of Theorem 3, let $\pi = \pi^\star$, and use Corollary 6 instead of Corollary 5 and the fact that $V_\mu^\pi \geq 0$. Then, we obtain desired result. $\square$

## C.5 Proof of Theorem 4

Define $U_k^\pi = V_\mu^\pi - V_\mu^{\pi_k}$, and the per-iteration distribution mismatch coefficient

$$\vartheta_k^\pi := \left\| \frac{\delta_\mu^\pi}{\delta_\mu^{\pi_k}} \right\|_\infty.$$

We first prove following key lemma.

**Lemma 13.** *Under Assumption 1, for a given $\mu$ with full support, the natural policy gradient with step size $\eta_k > 0$ generates a sequence of policies $\{\pi_k\}_{k=1}^\infty$ satisfying,*

$$\vartheta_{k+1}^\pi \left( U_{k+1}^\pi - U_k^\pi \right) + U_k^\pi \ \leq \ \frac{1}{\eta_k} KL_{\delta_\mu^\pi}(\pi, \pi_k) - \frac{1}{\eta_k} KL_{\delta_\mu^\pi}(\pi, \pi_{k+1})$$

*Proof of Lemma 13.* In the previous proof of Theorem 3, we showed that

$$\sum_{s \in \mathcal{S}} \delta_\mu^\pi(s) \sum_{a \in \mathcal{A}} Q^{\pi_k}(a, s)(\pi_k(a \mid s) - \pi_{k+1}(a \mid s)) + \sum_{s \in \mathcal{S}} \delta_\mu^\pi(s) \sum_{a \in \mathcal{A}} Q^{\pi_k}(a, s)(\pi(a \mid s) - \pi_k(a \mid s))$$

$$\leq \ \frac{1}{\eta_k} KL_{\delta_\mu^\pi}(\pi, \pi_k) - \frac{1}{\eta_k} KL_{\delta_\mu^\pi}(\pi, \pi_{k+1}).$$

For the first term

$$\sum_{s \in \mathcal{S}} \delta_\mu^\pi(s) \sum_{a \in \mathcal{A}} Q^{\pi_k}(a, s)(\pi_k(a \mid s) - \pi_{k+1}(a \mid s))$$

$$= \sum_{s \in \mathcal{S}} \frac{\delta_\mu^\pi}{\delta_\mu^{\pi_{k+1}}} \delta_\mu^{\pi_{k+1}} \sum_{a \in \mathcal{A}} Q^{\pi_k}(a, s)(\pi_k(a \mid s) - \pi_{k+1}(a \mid s))$$

$$\geq \left\| \frac{\delta_\mu^\pi}{\delta_\mu^{\pi_{k+1}}} \right\|_\infty \sum_{s \in \mathcal{S}} \delta_\mu^{\pi_{k+1}} \sum_{a \in \mathcal{A}} Q^{\pi_k}(a, s)(\pi_k(a \mid s) - \pi_{k+1}(a \mid s))$$

$$= \left\| \frac{\delta_\mu^\pi}{\delta_\mu^{\pi_{k+1}}} \right\|_\infty \left( V_\mu^{\pi_k} - V_\mu^{\pi_{k+1}} \right),$$

where the first inequality is from Lemma 7, and the last equality comes from Corollary 5.

For the second term, by Corollary 5,

$$\sum_{s \in \mathcal{S}} \delta_\mu^\pi(s) \sum_{a \in \mathcal{A}} Q^{\pi_k}(a, s)(\pi(a \mid s) - \pi_k(a \mid s)) = V_\mu^\pi - V_\mu^{\pi_k}.$$

We obtain desired result after substitution.

$$\square$$

We are now ready to prove Theorem 4

*Proof.* Since $\delta_\mu^{\pi_k}(s) \geq \mu(s)$ for all $s \in \mathcal{S}$, $\vartheta_k^\pi \leq \vartheta_\mu^\pi$ for all $k \geq 0$. $U_{k+1}^\pi - U_k^\pi \leq 0$ for all $k \geq 0$ by Lemma 12, and by Lemma 13,

$$\vartheta_\mu^\pi \left( U_{k+1}^\pi - U_k^\pi \right) + U_k^\pi \ \leq \ \frac{1}{\eta_k} KL_{\delta_\mu^\pi}(\pi, \pi_k) - \frac{1}{\eta_k} KL_{\delta_\mu^\pi}(\pi, \pi_{k+1}).$$

Dividing both sides by $\vartheta_\mu^\pi$ and rearranging terms, we obtain

$$U_{k+1}^\pi + \frac{1}{\eta_k \vartheta_\mu^\pi} KL_{\delta_\mu^\pi}(\pi, \pi_{k+1}) \ \leq \ \left( 1 - \frac{1}{\vartheta_\mu^\pi} \right) \left( U_k^\pi + \frac{1}{\eta_k(\vartheta_\mu^\pi - 1)} KL_{\delta_\mu^\pi}(\pi, \pi_k) \right).$$

Since the step sizes satisfy condition, $\eta_{k+1}(\vartheta_\mu^\pi - 1) \geq \eta_k \vartheta_\mu^\pi > 0$, we have

$$U_{k+1}^\pi + \frac{1}{\eta_{k+1}(\vartheta_\mu^\pi - 1)} KL_{\delta_\mu^\pi}(\pi, \pi_{k+1}) \ \leq \ \left( 1 - \frac{1}{\vartheta_\mu^\pi} \right) \left( U_k^\pi + \frac{1}{\eta_k(\vartheta_\mu^\pi - 1)} KL_{\delta_\mu^\pi}(\pi, \pi_k) \right).$$

Therefore, by recursion,

$$U_k^\pi + \frac{1}{\eta_k(\vartheta_\mu^\pi - 1)} KL_{\delta_\mu^\pi}(\pi, \pi_k) \ \leq \ \left( 1 - \frac{1}{\vartheta_\mu^\pi} \right)^k \left( U_0^\pi + \frac{1}{\eta_0(\vartheta_\mu^\pi - 1)} KL_{\delta_\mu^\pi}(\pi, \pi_0) \right).$$

$$\square$$

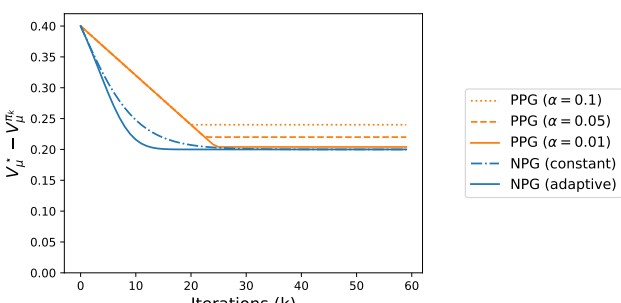

Figure 3: Comparison of projected policy gradient (PPG) and natural policy gradient (NPG) algorithms in Pathological MDP. Due to a discontinuity at optimal policy, $V_\mu^\star - V_\mu^{\pi_k} \geq V_\mu^\star - V_{+,\mu}^\star > 0$.

### C.6 Proof of Corollary 3

*Proof.* In the previous proof of Theorem 4, let $\pi = \pi^\star$ and use Corollary 6 instead of Corollary 5. Then, we obtain desired result. $\square$

## D Environments and additional experiment

### D.1 Environments

**Frozenlake** The Frozenlake environment is a $4 \times 4$ grid world consisting of a goal state, three terminal states, and frozen states. The agent has four actions: UP(0), RIGHT(1), DOWN(2), and LEFT(3). If the agent is in a frozen state, the environment executes the left/forward/right variants of the intended action with probabilities $1/3$ each. The agent receives a reward of $1$ only if it reaches the goal state. If the agent attempts to move off the grid, it stays in place.

**Cliffwalk** The Cliffwalk is a $3 \times 7$ grid world. The bottom right corner is the terminal goal state, and the states in the third row, except for the two end states, are terminal states. The agent has four actions: UP (0), RIGHT (1), DOWN (2), and LEFT (3). The MDP is deterministic, and the agent receives a reward of $1$ only when it reaches the goal state. If the agent attempts to move off the grid, it stays in place.

### D.2 Experiment on Pathological MDP

We run the projected policy gradient algorithm with $\alpha \in \{0.1, 0.05, 0.01\}$ and the natural policy gradient algorithm with both constant and adaptive step sizes. All algorithms are implemented using the transient policy gradient with the transient visitation measure. For Pathological MDP in Figure 1, we use $\{0.1 \cdot 1.01^k\}_{k=0}^\infty$ for the adaptive step size of natural policy gradient, where $k$ is the number of iterations, and $0.1$ for others.

The results are shown in Figure 3. As the figure shows, the policy error $V_\mu^\star - V_\mu^{\pi_k}$ remains strictly positive due to a discontinuity at optimal policy. $V^\star(s_1) = 0 > -1 = V^\pi(s_1)$ for any nonoptimal policy $\pi$ as discussed in Section 3. Thus, the iterates produced by policy gradient methods do not converge to the optimal value $V_\mu^\star$.

The natural policy gradient with adaptive step size still exhibits the fastest convergence rate among the algorithms, as the guaranteed linear rate of Theorem 4 predicts. Note that both natural policy gradients converge to $V_{+,\mu}^\star$ while the projected policy gradient converges to $V_\mu^{\pi_\alpha^\star}$ for each $\alpha$, and smaller $\alpha$ makes projected policy gradient converge closer to $V_{+,\mu}^\star$ since $V_\mu^{\pi_\alpha^\star}$ increases monotonically to $V_{+,\mu}^\star$ as $\alpha \to 0$.

