# OpenReview forum: "Why Policy Gradient Algorithms Work for Undiscounted Total-Reward MDPs"
_ICLR.cc/2026/Conference — Submitted to ICLR 2026_

### Official Review · Reviewer_LNyM · 2025-10-29

**Soundness:** 1
**Presentation:** 2
**Contribution:** 2
**Rating:** 2
**Confidence:** 4

**Summary:**

The paper considers the infinite-horizon and undiscounted Markov Decision Processes and studies the convergence of policy gradient theorem. It starts by first constructing a Pathological MDP and points out two discrepancies: the discontinuity of the value functions and the non-optimality of policies. The paper then introduces the definition of transient and recurrent states and divide the state space into two corresponding parts. Based on this, the paper introduces the transient visitation measure and the associated Transient Policy Gradient, followed by the convergence analysis and empirical results.

**Strengths:**

The paper is generally understandable and well written. The proposed problem of convergence analysis on policy gradient theorem is interesting when the policy gradients are applied to train large language models. The analysis is complete as it covers both theoretical results and empirical evaluation.

**Weaknesses:**

The paper in its current version has several major weaknesses that affects my assessment: 1. There are quite some vague (appears contradictory or even wrong) analysis and unstated assumptions that significantly downplays the potential significance of the results; 2. The empirical results appear not to reflect or corroborate the theoretical results and analysis.

### Confusing (and contradictory) analysis
1. In pathology 1, the discussion of discontinuity is confusing. “the optimal action at state $s_1$ is to remain at $s_1$”: (it can be true if the optimal policy for state $s_1$ is restricted to the set of deterministic policies $\Pi/\Pi_+$) then “any policy assigning a non-zero probability to the other action” (now the set of stochastic policies $\Pi_+$ is considered). It’s not clear if the discontinuity discussion is valid since two different sets are considered here. Importantly, if the discussions are narrowed to the set of stochastic policies, any stochastic policy at state $s_1$ would be optimal then.

2. In pathology 1, “a policy gradient method cannot be expected to succeed in the presence of such discontinuities”: the policy gradient method was proposed specifically to circumvent the discontinuities between deterministic policies and value functions, and hence to optimize stochastic policies only, see [Policy Gradient Methods for
Reinforcement Learning with Function Approximation]( https://proceedings.neurips.cc/paper_files/paper/1999/file/464d828b85b0bed98e80ade0a5c43b0f-Paper.pdf)

3. In pathology 2, “the optimal action at $s_2$ is to transition to $s_4$”: this might be wrong. Particularly, according to the analysis in Pathology 1, “any policy assigning a non-zero probability to the other action” would yield the same value and thus any stochastic policies $\Pi_+$ are optimal. But the follow-up discussion then restricts the policy set to be deterministic: $\pi(s)\in\arg\max Q^*(s, a)$. Evidently, a deterministic policy can’t be an optimal policy if the optimal policy belongs to the stochastic set. One example the Rock-Paper-Scissors game: any deterministic policy wouldn’t be optimal if the opponent is playing uniformly randomly.

4. Can authors discuss whether the transient visitation measure itself can be well-defined and under what types of Markov chains the Transient Policy Gradient should be applied upon? As the number of visits for these transient states is finite, compared to recurrent ones, the probability in Equation on line 261 can be all zeros. Consider the example in Figure 1, the transient visitation measure for state $s_2$ may not be defined: if the policy at state $s_2$ is to stay at $s_2$, then $s_2$ is recurrent; if the policy is to transition to $s_4$, then what’s the Probability of being at $s_2$? Furthermore, Transient Policy Gradient (Theorem 1) differs from the policy gradient theorem in that the state visitation measure has been replaced by the transient visitation measure. The original policy gradient theorem assumes the induced Markov chain is ergodic such that the state visitation measure is stationary (thus well-defined). But it’s unclear what Markov chain the Transient Policy Gradient is based upon. From reviewer’s perspective, it seems such transient policy gradient itself may be ill-defined.

5. The analysis of Natural Policy Gradient appears to repeat the existing results reported in Agarwal et al. (2021); Xiao (2022); Bhandari & Russo (2024); Mei et al. (2020), especially on the linear convergence with adaptive size. The only difference is the distribution mismatch coefficient. The closed form for the update rule does not provide any new insights. Further, the definition of modified Fisher information matrix can be problematic as $\delta_\mu^{\pi}$ may not be a well-defined probability distribution.

### Empirical evaluation

It’s unclear if the two toy examples, Frozenlake and CliffWalk, were used as undiscounted total-reward infinite-horizon MDPs (how to implement this as infinite-horizon MDPs? Is there any time-out used in the training?) Moreover, these two examples may not even have the transient states as defined by the paper (or the paper should highlight some of the states are transient?) The experiments on Pathological MDP in the appendix seem not to truly reach the conclusion that “it is not subject to this issue” (Line 172). The Transient Policy Gradient still suffers from the same issue?

**Questions:**

1. Line 034, “arbitrarily long horizons”, both references are actually considering trajectories with finite horizons (or truncated long trajectories into segments)

2. Line 043, “may ill-defined” --> “may be ill-defined”

3. Line 107, Can authors elaborate on how “the monotone convergence theorem guarantees that each limit exists (possibly infinite)”?

4. Line 132, the equation $Q^\pi = PV^\pi + r$ is not defined.

5. Line 136, can authors put the Theorem 7.19 for reference? “an optimal policy always exist in the undiscounted total-reward setup with finite state and action spaces”. The reviewer couldn’t find the edition of Puterman 2014. The online available edition is Puterman is 1994, 2005 by John Wiley & Sons.

6. Line 142, “$P^\pi(s)$ is defined as…” there is no such $P^\pi(s)$ defined (assume this is a typo);  Further, $P^\pi(s\rightarrow s’)$ is defined on a single sampled action $a\sim\pi$, not the expectation form? Also, $Pr$ is not defined, at least different from the transition matrix $P$.

7. Line 143, can authors elaborate the formula $V^\pi = \sum_{n=0}^{\infty} (P^\pi)^n r^\pi$? How to get this formula?

8. Line 185, “$\sum_{k=0}^{\infty} (P^\pi)^k(s, s) = \infty$: $(s, s)$ is not defined. Not sure if it refers to $P^\pi(s\rightarrow s)$.

9. Line 191, both $\bar{S}^\pi$ and $\bar{S}^\pi_n$ are not defined.

10. Line 208, the “reward-to-go” is not defined (or introduced beforehand).

11. Line 213, “$P^\pi(s_1, s_2)$” is not defined (or it refers to $P^\pi(s_1 \rightarrow s_2)$?)

12. Line 255, “the state visitation measure is defined as $d_{s_0}^\pi(s) = (1-\gamma)\sum_{i=0}^{\infty} (\gamma P^\pi)^I (s_0, s_i)$. The definition appears incorrect: 1. There is no $s$ on the right hand side; 2. There is no term of the initial state distribution $\mu$

13. Line 257, “In the undiscounted setting with $\gamma=1$, this object becomes undefined”. Even for undiscounted case, the state visitation measure may still be defined: check Eq. (8) in [Breaking the Curse of Horizon: Infinite-Horizon Off-Policy Estimation]( https://papers.nips.cc/paper_files/paper/2018/hash/dda04f9d634145a9c68d5dfe53b21272-Abstract.html)

14. Line 303, $\eta_k$ is not defined.

15. Line 370, $Q^{\pi_n}$ is not defined (or a typo?)

---

> ### Author Response · Authors · 2025-11-21
>
> We thank the reviewer for the constructive feedback. We are happy to hear that the reviewer found our results interesting. We address the reviewer's individual concerns in the following.
>
>
> **Weakness 1,2,3**: We apologize for the confusion. To clarify, our notion of 'discontinuity' refers to the discontinuity of the value function defined over the set of all (deterministic and stochastic) policies $\Pi$. (The set of 'deterministic' policies is a discontinuous set, but that is not what we are talking about.)
>
> In the classical discounted-reward setting, no such discontinuity arises, and we have the equalities
> $$
> \sup_{\pi\in \text{set of deterministic policies}} V^\pi(s)
> = \sup_{\pi\in \Pi \setminus \Pi_{+}} V^\pi(s)
> = \sup_{\pi\in \Pi} V^\pi(s)
> = \sup_{\pi\in \Pi_{+}} V^\pi(s),
> $$
> where $\Pi_{+}$ denotes the set of stochastic policies with full support (all actions have positive probability). This justifies the use of policy-gradient methods with softmax parameterizations, which by construction search only over  $\Pi_{+}$ as discussed in [1].
>
>
> In the undiscounted total-reward setting, however, the value function can be discontinuous even under Assumption 1 and even with finite state and action spaces. Our pathological MDP illustrates this with:
> $$
> 0 = \sup_{\pi\in \Pi} V^\pi(s_1)> \sup_{\pi\in \Pi_{+}} V^\pi(s_1) = -1.
> $$
> Thus, a policy-gradient method restricted to searching over $\Pi_{+}$ cannot approach the optimal deterministic policy. This is precisely what we meant by "a policy-gradient method cannot be expected to succeed in the presence of such discontinuities."
>
> Furthermore, we clarify that a policy $\pi$ is considered optimal if its value attains the maximum over all (deterministic and stochastic) policies $\Pi$, not just over the stochastic policies $\Pi_+$. As in the discounted total-reward setting, it is known that a deterministic optimal policy exists in our setting ([2, Theorem 7.1.9]). The example of Rock–Paper–Scissors raised by the reviewer has only a stochastic optimal policy, but that is a two-player game. Our setting concerns single-agent reinforcement learning, where this issue does not arise.
>
>
> **Weakness 4**: For any Markov chain with finite state and action spaces, the transient visitation measure is always well defined, and the transient policy gradient theorem holds under Assumption 1.
>
>
> By definition, the transient visitation measure assigns to each state a value greater than or equal to $1$, and as we emphasized in the main text, it is not a probability measure. In the pathological MDP, for example, if the policy at state $s_2$ transitions directly to $s_4$, then the transient visitation measure at $s_2$ is $1$.
>
>
> As the reviewer notes, [1] assumes ergodicity of the MDP to derive the policy gradient theorem using the stationary visitation measure. However, as clarified in [1], this assumption is for the *average-reward* setting, which is different from the *undiscounted total-reward* setting we study. To establish the policy gradient theorem in our setting, we rely on our novel insight: the invariance of recurrent and transient states on $\Pi_+$, constructed using the recurrent–transient theory. This insight enables us to define the transient visitation measure and prove the transient policy gradient theorem *without* imposing structural assumptions on the transition matrix.
>
> **Weakness 5**: In the appendix, we provide a mathematical derivation of the natural policy gradient formulation using the modified Fisher information matrix, and the derivation shows that the modified Fisher information matrix is well defined. For discussion of adapting prior analyses on the discounted reward setup to our undiscounted total reward setup,  please refer to our common response.
>
>
>
> **Weakness 6**: Yes, we used FrozenLake and CliffWalk as undiscounted total-reward infinite-horizon MDPs. As clarified in the appendix, both MDPs have terminal states, and every non-terminal state is transient for $\pi \in \Pi_+$. In our implementation, we used the explicit formulations of policy gradients in Sections 5 and 6, and since we consider the tabular case, we can avoid infinitely long simulations or timeouts through calculations involving matrix inverses.
>
>
> Regarding the statement that "it is not subject to this issue," we were referring specifically to *pathology 2*. However, our experiment on the pathological MDP is demonstrating *pathology 1*.

---

> ### Author Response · Authors · 2025-11-21
>
> **Question 1**: In practice, trajectories must be finite due to practical limitations. For example, when training an LLM with RL, trajectories are truncated because of maximum token limits. However, this restriction comes from practical constraints. We argue that it is still valuable to study the infinite-horizon MDP as an idealized model, with finite-length trajectories viewed as truncations or approximations of this infinite-horizon process.
>
>
> **Question 2**:  Thank you for pointing out this typo. We have corrected in the revised manuscript.
>
>
>
> **Questions 3,7**
> In the definition of $V^\pi\_{+}$, we see that $\max\\{r,0\\}$ is always nonnegative. So the monotone convergence theorem guarantees the existence of the limit
> $$
> \lim\_{T \rightarrow \infty}
> \mathbb{E}\_\pi\bigg[\sum^{T-1}\_{i=0} \max\{r(s_i,a_i),0\}
>   \,\Big|\, s\_{0}=s\bigg]
> = V_{+}^\pi(s).
> $$
> By the same argument, $V\_{-}^\pi(s)$ is also well defined. Finally, under Assumption~1,
> $$
> V^\pi(s) = \lim\_{T \rightarrow \infty} \mathbb{E}\_\pi\bigg[\sum^{T-1}\_{i=0} r(s_i,a_i)  \,\Big|\, s\_{0}=s\bigg]
> $$
> is well defined. By computing this expectation in closed form, we obtain
> $$
> V^\pi = \sum_{n=0}^{\infty} (P^\pi)^n r^\pi.
> $$
>
>
>
> **Question 4**: Sorry for the confusion. In $Q^\pi = P V^\pi + r$, the $P$ represents the transition matrix in $\mathbb{R}^{|\mathcal{S}||\mathcal{A}| \times |\mathcal{S}|}$, and $P V^\pi$ denotes the matrix-vector product. We properly defined this in the revised manuscript.
>
> **Questions 5**: Sorry for the confusion. This was a typo, and the correct numbering is "Theorem 7.1.9'. [Puterman  2005] also uses the same numbering.
>
> **Questions 6,8,11**: Sorry for the confusion. We now explicitly define $\mathcal{P}^\pi(s,s')$, and use $\mathrm{Prob}$ to denote probability. With the clarified definition of $\mathcal{P}^\pi$, we can regard $\mathcal{P}^\pi$ as an $|\mathcal{S}|\times|\mathcal{S}|$ matrix and $(\mathcal{P}^\pi)^n$ as a matrix as well.
>
>
> **Question 9**" Sorry for the confusion. $\bar{S}^\pi = (\bar{S}^\pi)^1$, and $(\bar{S}^\pi)^n$ indicates the transition probability from transient states to recurrent states in $(P^\pi)^n$. We have clarified this in the revised manuscript.
>
>
> **Question 10**: Sorry for the confusion. We have corrected the wording to "reward".
>
> **Question 12**: Sorry for the confusion. We changed the definition of the state visitation measure to be consistent with the transient visitation measure. Please check our revised version.
>
> **Question 13**: Yes, but again, [3] also considers the average-cost MDP setup, which is different from our setup.
>
> **Question 14**: Sorry for the confusion; $\eta_k$ denotes the stepsize of the $k$-th iterate.
>
> **Question 15**: Thank you for pointing out the typo. Yes, $Q^{\pi_n}$ should be changed to $Q^{\pi_i}$.
>
> Given that we positively address the main concerns of the reviewer, we kindly ask the reviewer to consider raising the score.
>
> References
>
> [1] Sutton, R. S., McAllester, D., Singh, S., $\\&$ Mansour, Y. (1999). Policy gradient methods for reinforcement learning with function approximation. Advances in neural information processing systems
>
> [2] Puterman, M. L. (2014). Markov Decision Processes: Discrete Stochastic Dynamic Programming. John Wiley and Sons, 2nd edition
>
> [3] Liu, Q., Li, L., Tang, Z., $\\&$ Zhou, D. (2018). Breaking the curse of horizon: Infinite-horizon off-policy estimation. Advances in neural information processing systems

---

### Official Review · Reviewer_aaNs · 2025-10-30

**Soundness:** 3
**Presentation:** 3
**Contribution:** 3
**Rating:** 4
**Confidence:** 3

**Summary:**

This paper studies the convergence of policy gradient methods in the undiscounted total-reward infinite-horizon setting. Recent applications including RLHF and RLVR for LLMs often use undiscounted total reward settings, motivating theoretical foundations in this regime. The paper introduces a transient visitation measure to replace the standard discounted visitation measure and uses a recurrent-transient decomposition of the MDP to derive convergence results for projected and natural policy gradient algorithms.

**Strengths:**

- The paper develops a rigorous convergence analysis for policy gradient methods in an underexplored setting, providing a new transient policy gradient theorem
- The recurrent–transient decomposition is carefully applied, leading to clean lemmas (Lemma 2, Lemma 5) and new insight into continuity properties of the value function
- Clear presentation in terms of the gradual development from Section 3 to Section 5.

**Weaknesses:**

- Assumption 1 (finite total reward criterion) is quite strong and misleading in its current form of presentation. Assumption 1 enforces finiteness of value functions for all, which implies all recurrent states must have zero reward (Lemma 1). This effectively removes any interesting behavior in recurrent states and turns the problem into one over transient MDPs. This should not be buried as an assumption as it fundamentally alters the nature of the problem. The title and abstract should make this restriction explicit.
- I believe the paper has limited technical novelty because once the transient-state formulation and restrictions are in place, the resulting analysis mainly requires continuity and contraction properties and the rest of the proof mirrors existing proof techniques. The “transient visitation measure” is essentially an adaptation of the standard visitation measure to a subspace.
- If transient states are visited only finitely often, then it is unclear how one can meaningfully learn about them from sample-based estimates. The current theoretical setup assumes full knowledge of transitions and gradients, but practical implications for sample-based or online algorithms remain entirely open.

**Questions:**

Suggestions for Clarity:
- The paper motivates its setting using language modeling and RLHF, claiming these are undiscounted total-reward infinite-horizon problems. However, in practice, both are finite-horizon due to maximum token limits. The authors should clarify why it makes sense to model such tasks as infinite-horizon.
- The paper contrasts its setting with discounted MDPs but does not sufficiently discuss its relationship to average-reward (multi-chain) MDPs and the literature therein (Putterman, 1994), which are the more natural analogues of undiscounted problems. The distinctions between total and average reward formulations should be made explicit.

---

> ### Author Response · Authors · 2025-11-21
>
> We thank the reviewer for the thoughtful feedback. We are happy to hear that the reviewer found our results novel and clean. We address the reviewer's individual concerns in the following.
>
>
> **Weakness 1**: As the reviewer correctly notes, Assumption 1 is indeed a strong assumption and distinguishes our setting from prior work on the stochastic shortest path problem and the positive/negative models.
>
> As our title (“Why Policy Gradient Works for Undiscounted Total-Reward MDPs”) suggests, our focus is on understanding when policy gradient methods function correctly in the undiscounted total-reward setting. Under that rubric, identifying Assumption 1, finiteness of the value function, as a necessary condition for policy gradient to succeed is a contribution, providing a point of clarity. Moreover, as stated in the abstract, our formulation is motivated by current RL frameworks for large language models, and Assumption 1 naturally arises from this application.
>
>
>
>
> **Weakness 2**: In our view, the fact that our formulation via the transient visitation measure allows the prior proof techniques to be straightforwardly adapted to our setup is a strength, not a weakness of our work. Please refer to the common response.
>
>
>
>
>
> **Weakness 3**: The reviewer raised a good point.
> A plan we have for follow-up work is to consider the stochastic setup with a REINFORCE-style formulation [1] to compute the stochastic policy gradient. Roughly speaking, for a given initial distribution $\mu$, policy $\pi$, and trajectory $\tau = (s_0,a_0,s_1,a_1,\dots)$, we can consider the trajectory distribution
> $$
> \text{Prob}^{\pi}_{\mu} (\tau)= \mu(s_0) \prod\_{i=0}^{\infty} \pi( a_i \mid s_i)P(s\_{i+1}\,|\,s_i,a_i).
> $$
> Defining the total reward of a trajectory as
> $R(\tau)=\sum\_{i=0}^{\infty}r(s_i, a_i)$, we have
> $$
>  V^{\pi}\_{\mu}=\mathbb{E}\_{\tau \sim \text{Prob}^{\pi}\_\mu (\cdot)}[R(\tau)].
> $$
> Then, we can obtain
> $$
> \nabla\_{\theta} V^{\pi\_{\theta}}\_{\mu}=\mathbb{E}\_{\tau \sim \text{Prob}^{\pi\_\theta}\_\mu (\cdot)}\left[ R(\tau) \left(\sum\_{t=0}^{\infty}\nabla\_{\theta}\log \pi\_{\theta}(a_t \mid s_t) \right)\right],
> $$
> and it is worth mentioning that Assumption 1 will be needed for this derivation to be mathematically consistent. Finally, based on this formulation, we can consider a Monte Carlo estimator of the policy gradient, which can be used to implement sample-based or online algorithms in the undiscounted total-reward  setup.
>
>
>
> **Question 1**: As the reviewer points out, in practice, trajectories must be finite due to practical limitations. For example, when training an LLM with RL, trajectories are truncated because of maximum token limits. However, this restriction comes from practical constraints. We argue that it is still valuable to study the infinite-horizon MDP as an idealized model, with finite-length trajectories viewed as truncations or approximations of this infinite-horizon process.
>
>
> **Question 2**: Thank you for the comment. Following the reviewer’s suggestion, we elaborated on the distinction between the average-cost setup and our undiscounted total-reward setup in the related works section of the revised manuscript.
>
>
> Given that we positively address the main concerns of the reviewer, we kindly ask the reviewer to consider raising the score.
>
>
>
> References
>
>
> [1] Williams, R. J. (1992). Simple statistical gradient-following algorithms for connectionist reinforcement learning. Machine learning, 8(3), 229-256.

---

> > ### Comment · Reviewer_aaNs · 2025-11-21
> >
> > I thank the authors for their response and maintain my score as I continue to believe that the paper has limited technical novelty.

---

### Official Review · Reviewer_F2er · 2025-10-31

**Soundness:** 1
**Presentation:** 1
**Contribution:** 1
**Rating:** 0
**Confidence:** 5

**Summary:**

The paper studies policy gradient methods in the undiscounted infinite-horizon setting. However, the authors overlook the extensive prior literature on average-cost MDPs, where this problem has been thoroughly analyzed since the 1990s. Conceptually, the formulation is fundamentally flawed: the paper considers an undiscounted total-reward criterion over an infinite horizon without any normalization or averaging, which leads to divergence unless the rewards are zero. The ad hoc assumption that both the positive and negative parts of the value function remain finite does not resolve this inconsistency. Beyond these foundational issues, the paper offers little novelty or insight, and the technical assumptions undermine its validity. Overall, the contribution is not meaningful in its current form, and I recommend rejection.

**Strengths:**

None.

**Weaknesses:**

- The paper overlooks the extensive literature on average-cost MDPs, where the undiscounted infinite-horizon problem has already been analyzed in depth from multiple theoretical perspectives.

- The assumption that the value function remains finite without discounting or averaging is unjustified and mathematically inconsistent under standard reward structures. The analysis ignores the fundamental role of discounting or averaging in ensuring convergence of value functions, leaving the proposed framework theoretically unsupported.

**Questions:**

Have you considered that Assumption 1 can only hold if the reward function is identically zero within the recurrent class?

---

> ### Author Response · Authors · 2025-11-21
>
> We firmly disagree with the reviewer's claim (i) that we overlook the prior literature on average-cost MDPs and (ii) that our assumption is mathematically inconsistent.
>
>
> **(i)** We clarify that the undiscounted average-cost MDP is a very different setting from the undiscounted total-reward MDP we study. This is not a matter of the authors overlooking prior work; we are familiar with the literature on average-cost MDPs and have prior experience working in that area.
>
> Regarding relevant prior work, we do acknowledge the stochastic shortest path problem  [1] and the positive/negative models for total-reward MDPs [2, Sections 7.2–7.3].
>
> Could the reviewer clarify which specific prior results on average-cost MDPs are relevant to our work?
>
>
> **(ii)** As the prior literature [1,2] already establishes, the undiscounted total-reward setup is mathematically consistent. Under Assumption 1, our Lemma 1 shows that the reward function must be identically zero at recurrent states, which indeed imposes a specific reward structure the reviewer points out. As we discuss in the main text, similar structural conditions and finiteness results for the value function have been analyzed in earlier works [1–3]. In our work, we find the motivation for this assumption in current applications of RL to LLMs, where positive reward is only given at the end when the LLM produces the correct sequence of tokens. The terminal state can then be viewed as a recurrent state with zero reward, so Assumption 1 is satisfied under this reward structure.
>
> We do not see any unjustified or mathematically inconsistent aspects of our setup. Could the reviewer elaborate on which parts they believe are unjustified or are mathematically inconsistent?
>
>
> References
>
> [1] Bertsekas, D. P., $\\&$ Tsitsiklis, J. N. (1991). An analysis of stochastic shortest path problems. Mathematics of Operations Research, 16(3), 580-595
>
> [2] Puterman, M. L. (2014). Markov Decision Processes: Discrete Stochastic Dynamic Programming. John Wiley and Sons, 2nd edition
>
> [3] Schäl, M. (1983). Stationary policies in dynamic programming models under compactness assumptions. Mathematics of Operations Research, 8(3), 366-372

---

### Official Review · Reviewer_2pSR · 2025-11-01

**Soundness:** 4
**Presentation:** 4
**Contribution:** 3
**Rating:** 8
**Confidence:** 4

**Summary:**

This paper studies policy gradient methods for undiscounted total reward infinite-horizon MDPs. After introducing the setting, a few differences with the discounted setting are discussed in section 3. Then, section 4 established a policy gradient theorem using some insights from the theory of Markov chains, introducing in particular a transient visitation measure.  It is also shown that the recurrent-transient classification of states is independent of the choice of the policy (in the set of policies with full support). Section 5 provides a sublinear ($1/k$ order) convergence rate for projected policy gradient descent when the projection is over a set of full support policies (controlled by a constant $\alpha$). Proofs rely on showing smoothness of the value function combined with a gradient domination property similarly to the undiscounted setting. Finally, section 6 analyzes natural policy gradient with a constant or an adaptive step size showing sublinear and linear rates respectively. A simple toy experiment concludes the paper by illustrating the convergence behavior of the different proposed methods.

**Strengths:**

1. The paper is nicely written and to the point. The presentation is clear and the paper is well organized, making the paper easy to read (at least for someone familiar with the theory of policy gradient methods). I enjoyed reading it.
2. The contributions are solid and novelty is clear, building on a few insights from the fundamental theory of (finite state) Markov chains (recurrence and transience). The simplification of Lemma 1, 2 which also leads to the policy gradient theorem is a nice insight (for undiscounted infinite-horizon MDPs). The results are sound, I skimmed through the proofs which look correct to the best of my knowledge (not line by line though for some of them), they mostly follow similar lines as prior work.
3. The literature on the theory of policy gradient methods for discounted infinite-horizon MDPs has witnessed a lot of developments in the last few years. However, less is known for undiscounted infinite horizon settings. This paper provides a nice and solid addition to the literature, establishing similar results as the discounted setting with some nuances and caveats.
4. The relevance of the paper is further supported by the fact that in practice, discounting is not always used (probably even seldom used) as mentioned in the paper. I believe this is important and will open the door to further developments. This also complements a recent trend in the development of guarantees beyond the discounted settings, e.g. last year at the same conference with the study of policy gradient methods for the average reward setting.

**Weaknesses:**

1. **Technical novelty**: the development of the results mainly follows existent techniques (Xiao 2022 in particular) with a few adaptations using the theory of Markov chains (for finite states) which are fairly straightforward (as the proofs p. 13 show for instance). Overall, I think that technical novelty is limited but this is not a major weakness in my opinion as there are some interesting distinctions with the discounted setting and the analysis is well executed.

2. **Corollary 1**: $\alpha$ depends on $\epsilon$. It is not clear how the constant $C_{\alpha}$ behaves as a function of $\epsilon$. Hence it is unclear if the ‘convergence rate is sublinear’ as stated in l. 348 in this setting.

3. **Policy gradient theorem:** It is not clear if the policy gradient theorem is operationalizable: can we actually compute stochastic policy gradients using the provided formula in a different way than usual? Is the transient kernel $T$ actually accessible? Is there a way to write the same theorem using trajectories like in the discounted setting? Or can one just also write the same policy gradient as usual for sampling and the theorem mainly serves as a tool for analysis. This somehow relates to the stochastic setting, but still I think it would be useful to comment on that since (a) the value of the policy gradient theorem is usually (partly) in the stochastic policy gradient computation it can unlock and (b) the paper does talk about policy gradient algorithms, so a word about their implementation would be welcome.

4. **Theorem 4**: I think it would be useful to comment on cases where the convergence rate is truly linear, i.e. cases where the distribution mismatch between the initial distribution and the visitation measure (at an optimal policy) is strictly larger than 1 (it might almost always be given the definition of the visitation measure). Giving more intuition on what makes convergence faster  would be useful since the theorem somehow provides an instance dependent rate (different from the other results in the paper).

**Typos:**
- l. 54: ‘stochastic’ capital letter.
- l. 97: $\pi(a|s)$ instead of $\pi(s|a)$.
- l. 131: $P^{\pi}$ instead of $P$?
- l. 191: the matrix $S$ is not defined.
- l. 286: $\pi_{\theta}$ instead of $\pi$ under the expectation.
- l. 309: stp, setup
- l. 357: $\theta_{s,a’}$ in the denominator.
- l. 361: ‘, The’, lower case.
- l. 370: $\pi_i$ for the Q-function term.
- Proof lemma 5, some undesirable commas remaining.
- l. 731: 'exsit', exist.
- l. 742, 743: $\pi_{\Pi}$?, 'diffretiable'.

**Questions:**

1. l. 157-161: can you highlight better the contrast with the discounted setting in terms of pathology?
2. l. 335: Can you justify the limit when $\alpha \to 0$? Does the constant $C_{\alpha}$ remain controlled as a function of $\alpha$ when $\alpha \to 0$?
3. l. 397: $V_{+,\mu}^{\star}$ is defined as a supremum. Is it attained at some policy? How do you obtain this limit statement? Do you achieve that the value is achieved at some policy $\pi^{\star}$ and plug in that policy in l. 395 using $\pi=\pi^{\star}$?
4. It is known in the discounted setting that NPG approximates Policy Iteration (PI) which is known to converge linearly using $\gamma$-contraction (as discussed e.g. in Xiao 2022). Is there any such meaningful link to draw here in the undiscounted setting with PI (without $\gamma$ contraction), for instance to interpret/discuss Theorem 4?
5. Corollary 2 (minor): Is there a missing norm of $V^{\pi_0}$ term in the bound since this is a corollary of Theorem 3?
6. Proof of Lemma 3, l. 671 (minor): Why does fixed transient class (invariance to policy) imply continuity of $T^{\pi}$? I guess it is useful to stress that the form of the matrix will stay the same but for continuity not sure it is enough, can't we just say that $P^{\pi}$ is continuous in $\pi$ (as it is linear in $\pi$) and then $\bar{T}^{\pi}$ is a submatrix of $P^{\pi}$ hence also continuous (as a fixed projection by the invariance shown)?

---

> ### Author Response · Authors · 2025-11-21
>
> We thank the reviewer for the extremely thorough feedback. We are happy to hear that the reviewer found our results novel and valuable. We address the reviewer's individual concerns in the following.
>
>
> **Weakness 1**: We agree with the reviewer's perspective and elaborated on this in the common response. Thank you for your comment.
>
> **Weakness 2 and Question 2**: Indeed, we expect $C_\alpha$ to grow as $\epsilon\rightarrow 0$, and therefore Corollary 1 does not provide an explicit iteration complexity as a function of $\epsilon$. Strictly speaking, however,  "the convergence rate of projected policy gradient is sublinear" is still correct: *sublinear convergence* does not assert an $O(1/\epsilon)$ iteration complexity, but rather indicates that the complexity is worse than $O(\log(1/\epsilon))$.
>
> Importantly, we do guarantee convergence in the sense that for any fixed $\epsilon > 0$, there exists a finite $k$ such that the algorithm achieves an $\epsilon$-optimal policy.
>
>
> **Weakness 3**:
> The reviewer raised a good point.
> A plan we have for follow-up work is to consider the stochastic setup with a REINFORCE-style formulation [1] to compute the stochastic policy gradient. Roughly speaking, for a given initial distribution $\mu$, policy $\pi$, and trajectory $\tau = (s_0,a_0,s_1,a_1,\dots)$, we can consider the trajectory distribution
> $$
> \text{Prob}^{\pi}_{\mu} (\tau)= \mu(s_0) \prod\_{i=0}^{\infty} \pi( a_i \mid s_i)P(s\_{i+1}\,|\,s_i,a_i).
> $$
> Defining the total reward of a trajectory as
> $R(\tau)=\sum\_{i=0}^{\infty}r(s_i, a_i)$, we have
> $$
>  V^{\pi}\_{\mu}=\mathbb{E}\_{\tau \sim \text{Prob}^{\pi}\_\mu (\cdot)}[R(\tau)].
> $$
> Then, we can obtain
> $$
> \nabla\_{\theta} V^{\pi\_{\theta}}\_{\mu}=\mathbb{E}\_{\tau \sim \text{Prob}^{\pi\_\theta}\_\mu (\cdot)}\left[ R(\tau) \left(\sum\_{t=0}^{\infty}\nabla\_{\theta}\log \pi\_{\theta}(a_t \mid s_t) \right)\right],
> $$
>  and it is worth mentioning that Assumption~1 will be needed for this derivation to be mathematically consistent. Finally, based on this formulation, we can consider a Monte Carlo estimator of the policy gradient, which can be used to implement sample-based or online algorithms in the undiscounted total-reward  setup.
>
>
> As the reviewer correctly pointed out, the transient visitation measure is not needed in the REINFORCE-style algorithm; it serves purely as an analytical tool, just as in the discounted case.
>
>
>
>
> **Weakness 4**:
> Thank you for the comment. If $\mu$ has full support, then the distribution mismatch $\vartheta^\pi_\mu$ is always finite by definition, and $1 - 1/\vartheta^\pi_\mu < 1$, which leads to linear convergence of natural policy gradient. Also, we can see that as the distribution mismatch becomes smaller, the linear convergence becomes faster. Lastly, the distribution mismatch is equal to one if and only if there exist no transient states, and in this case, the value function must be zero. As the reviewer suggested, we have elaborated this point in the revised version.

---

> ### Author Response · Authors · 2025-11-21
>
> **Typos**: Thank you for pointing out the typos. We have corrected them in the revised manuscript. We briefly note that, for line 131, $P$ denotes the transition matrix in $\mathbb{R}^{|\mathcal{S}||\mathcal{A}| \times |\mathcal{S}|}$, and $P V^\pi$ indicates the matrix-vector product. We clarified this in the revised version. Also, for line 191, sorry for the confusion. $\bar{S}^\pi$ denotes the transition probability from transient states to recurrent states in $P^\pi$. We have clarified this as well in the revised version.
>
> **Question 1**: Thank you for the suggestion. We added the phrase “while the discounted-reward setup guarantees differentiability of the value function” in the revised version.
>
>
> **Question 3**: Sorry for the confusion. To sketch the proof, for a fixed stepsize $\eta$ and given $\epsilon > 0$, there exists a policy $\pi$ such that $V^\star\_{+,\mu} - V^\pi\_\mu < \epsilon/2$, since $V^\star\_{+,\mu} < \infty$ under Assumption 1. Then, by the inequality in Theorem 3,, there exists $k$ such that $V^\pi - V^{\pi_k}\_\mu < \epsilon/2$. Therefore, we have $V^\star\_{+,\mu} - V^{\pi_k}\_\mu < \epsilon$, and since this holds for any $\epsilon > 0$, we obtain $V^{\pi_k}\_\mu \rightarrow V^\star\_{+,\mu}$. In the proof, we do not use or assume the existence of a policy $\pi$ such that $V^\pi\_\mu = V^\star\_{+,\mu}$. We elaborated this proof in the appendix of the revised version.
>
>
> **Question 4**: The reviewer raises an interesting point. As the reviewer pointed out, [2] showed that by letting $\eta_k \rightarrow \infty$, NPG reduces to policy iteration (PI), and they directly obtained the contractive convergence of PI from the convergence rate of NPG. However, the problem in applying this approach to our setup is that policy iteration considers deterministic policies, while our framework requires policies to have full support. Careful analysis would be needed, but from our perspective, because of this issue of support, it is unclear how to derive a convergence result for PI from the convergence result of NPG in our setup. Additionally, as far as we know, in the undiscounted total-reward setup, it is only known that policy iteration converges in a finite number of steps, and its convergence rate is not known [3, Theorem 7.2.6].
>
> **Question 5**:  Roughly speaking, in the proof of Corollary 2, the term
> $V^{\pi_k} - V^{\pi_0}$ appears, and under the nonnegative reward assumption, $V^{\pi_0} \ge 0$. Then, we have $V^{\pi_k} - V^{\pi_0} \le V^{\pi_k} \le V^{\pi_\star}$. This is why the $V^{\pi_0}$ terms disappear under the nonnegative reward assumption in Corollary 2.
>
> **Question 6**:  Thank you for the suggestion. Following the reviewer’s simple and compact argument, we revised the proof of Lemma 3. Thank you again for the detailed feedback.
>
> References
>
> [1] Williams, R. J. (1992). Simple statistical gradient-following algorithms for connectionist reinforcement learning. Machine learning, 8(3), 229-256
>
> [2] Xiao, L. (2022). On the convergence rates of policy gradient methods. Journal of Machine Learning Research
>
> [3] Puterman, M. L. (2014). Markov Decision Processes: Discrete Stochastic Dynamic Programming. John Wiley and Sons, 2nd edition

---

### Official Review · Reviewer_mXRp · 2025-11-10

**Soundness:** 3
**Presentation:** 3
**Contribution:** 2
**Rating:** 6
**Confidence:** 4

**Summary:**

The paper considers developing a policy gradient algorithm for total reward MDPs. The entire formulation and study is in the context of planning (ie no learning). Previous contexts of total reward MDPs study bounded non-negative, or negative value functions, or the stochastic shortest path problem where as the current paper considers all MDPs with a bounded value function (corresponding to both negative and non negative rewards).

The core intuition lies in identifying the role of transition states. Since the assumption of bounded value functions necessarily imply all recurrent states to have a reward of 0, the analysis boils down to studying the behavior of transient states. The authors heavily leverage the finiteness of states and actions in their analysis and the intuition that under policies that have nonzero measure on all actions have the same characterization of transient and recurrent states.

The authors then proceed to analyze global convergence of policy gradient (and provide sublinear rates), natural policy gradient (NPG) (provide a sublinear rate) and NPG with adaptive step size (provide a linear rate). They further conduct simulations to verify the theory.

**Strengths:**

The problem studied is of importance but has not received much attention due to some pathological conditions (such as lack of continuity of value function wrt policy and existence of solutions to the bellman equation). The authors provide a principled approach to this and analyze standard policy gradient algorithms in this setting.

The paper is over well written and the core intuition behind their analysis is easy to follow.

**Weaknesses:**

The technical novelty is not entirely clear. The analysis is pretty much exactly same as prior works except for considering the transient state probabilities instead of the entire probability transition matrix.

The policy gradient analysis seems moot since the NPG bounds seem to perform much better and have better convergence properties (no dependence on state and action spaces).

The results would be a lot more compelling if the state and actions spaces are countable (In practice many systems are incredibly large so analyzing those systems without relying on finiteness of state action spaces would make for a very interesting problem).

**Questions:**

Some typos: line 54- Stochastic, line 97: $\pi(a|s), line 309: step,  line 334 Theorem 1,

1. Does zero reward for recurrent states imply finiteness of value function?

2. In Lemma 5, isnt the $V^\pi(s)=0$ for all recurrent states true for all policies under assumption 1?

3. In corollary 1, what is the intuition behind that choice of $\alpha$?

4. Is the infinity norm in the finite rates in theorem only with respect to the transient states?(since $\delta$ is defined only for transient states)

5. How are equations in lines 363-366 related to the equation in line 370?

6. Why do we need $\mu$ to have a full support (over transient(?) states)? Shouldn't it suffice to start at some fixed transient state and the conditions for existence of solutions etc would still be met?

---

> ### Author Response · Authors · 2025-11-21
>
> We thank the reviewer for the insightful feedback. We are happy to hear that the reviewer found our results principled and clear. We address the reviewer's individual concerns in the following.
>
>
> **Weakness 1**: In our view, the fact that our formulation via the transient visitation measure allows the prior proof techniques to be straightforwardly adapted to our setup is a strength, not a weakness of our work. Please refer to the common response.
>
> **Weakness 2**: As the reviewer pointed out, the natural policy gradient exhibits a faster linear rate while the projected policy gradient exhibits a sublinear rate in the undiscounted total-reward setup, and similar rates also appear in the discounted reward case [1]. However, in the sampling setup of the discounted reward, [2] proved that the stochastic natural policy gradient might not converge, while the stochastic policy gradient with softmax parametrization converges in probability.
>
> If an analogous convergence result holds in the sampling version of our setup, then the projected policy gradient could exhibit better theoretical convergence guarantees than the natural policy gradient. Extending our analysis to the sampling-based setting is, therefore, an interesting direction for future work.
>
>
> **Weakness 3**: The reviewer raises an excellent point. We did consider such an extension, but the proof of Proposition 1, the invariance of recurrent and transient states, appears to be a key obstacle. Our proposition relies on a classical result in recurrent-transient theory: in finite-state MDPs, a communicating class is closed if and only if it is recurrent ([3, Theorem 3.2.8]). However, this equivalence fails in general for countably infinite state spaces, and known counterexamples prevent a straightforward extension.
>
> We attempted to address this by refining the classification of states more delicately and by exploring whether assumptions such as ergodicity or weakly communicating structures would suffice. With additional structural assumptions, we believe that our framework could indeed be extended to the continuous or countably infinite setting.
>
>
> **Question 0**: Thank you for pointing out these typos. We have corrected them in the revised manuscript.
>
> **Question 1**: The reviewer raised an interesting point. Yes, if the rewards at recurrent states are zero for a given policy $\pi$, then the value function is finite. Thus, Assumption 1 (finiteness of the value function) holds if and only if, for all policies $\pi$, the rewards induced by policy at recurrent states are zero.
>
>
> **Question 2**: As the reviewer points out, $V^\pi(s) = 0$ for a recurrent state $s$. But Lemma 5 is considering $V^{\pi'}(s) = 0$ with $s$ being a recurrent state of $\pi$, not $\pi'$.
>
>
> **Question 3**: The intuition behind the choice of $\alpha$ comes from our Lemma 5, the transient performance difference lemma. This lemma shows that the difference between two value functions can be bounded by the difference between two policies, the value function, and the transient visitation measure. Specifically, $Q^\star$ and $\delta^{\pi_\star}_\mu$ come from the right-hand side of the inequality in Corollary 1 under the non-negative reward assumption, and this leads to our specific choice of $\alpha$.
>
>
> **Question 4**: We clarify that our transient visitation measure is also defined for recurrent states. Specifically, by the definition in Section $4$, the transition matrix is set to zero at recurrent states, and thus for a recurrent state $s$ and initial state $\mu$, $\delta^\pi_\mu(s) =\mu(s)$. This directly implies that our theorems hold with respect to all states.
>
> **Question 5**: Sorry for the confusion. We explicitly write out the derivation in the appendix of the revised manuscript, and the derivation basically follows [5] in the discounted total-reward case.
>
> **Question 6**: The reviewer is very observant. Rigorously speaking, $\mu$ doesn't need to have full support. A necessary and sufficient condition for distribution mismatch coefficient $\||\frac{\delta\_{\mu}^{\pi_\star}}{\mu}\||\_{\infty}$ to be finite is that the support of $\mu$ contains the support of $\delta^{\pi_\star}_{\mu}$. However, for simplicity, we assume that $\mu$ has full support.
>
>
> Given that we positively address the main concern, we kindly ask the reviewer to consider raising the score.

---

> ### Author Response · Authors · 2025-11-21
>
> References
>
> [1] Xiao, L. (2022). On the convergence rates of policy gradient methods. Journal of Machine Learning Research
>
> [2] Mei, J., Dai, B., Xiao, C., Szepesvari, $\\&$ C., Schuurmans, D. (2021). Understanding the effect of stochasticity in policy optimization. Advances in Neural Information Processing Systems
>
> [3]  Brémaud, P. (2013). Markov chains: Gibbs fields, Monte Carlo simulation, and queues (Vol. 31). Springer Science $\\&$ Business Media
>
> [4] Puterman, M. L. (2014). Markov Decision Processes: Discrete Stochastic Dynamic Programming. John Wiley and Sons, 2nd edition
>
> [5] Agarwal, A., Kakade, S. M., Lee, J. D.,  $\\&$ Mahajan, G. (2021). On the theory of policy gradient methods: Optimality, approximation, and distribution shift. Journal of Machine Learning Research

---

### Author Response · Authors · 2025-11-21
**Common Response**

First of all, we thank the reviewers for their detailed and thoughtful feedback. The reviewers generally agree on the importance of the problem and the novelty of our core insight, while noting that the analyses in Sections 5 and 6 closely mirror proof strategies from prior work in the discounted-rewards setting.

Indeed, we show that by decomposing the states into recurrent and transient states and introducing the transient visitation measure, techniques developed for the discounted setting can be carried over to the undiscounted total-reward setup. We believe this is a valuable insight.

In our view, the fact that our novel reformulation allows proof techniques from a different setup to transfer cleanly into this one **is a strength, not a weakness**. Such reformulations make connections between subareas, and these connections are, we argue, more valuable than entirely separate analyses based on new and disjoint techniques.

Our intent was to present this main point in clear, to-the-point writing. We respectfully ask the reviewers to assess our work based on the value this core insight provides.

---

### Meta-Review · Area_Chair_t5v4 · 2026-01-07

**Summary:**

The reviews are highly divergent, with one reviewer finding the analysis well executed and others raising serious concerns about theoretical soundness, positioning, and novelty. While the rebuttal clarified several technical points and addressed many clarification questions, two core issues remain unresolved: (1) the limited and insufficiently articulated technical novelty relative to prior work, and (2) the lack of a meaningful discussion connecting the undiscounted total-reward setting to the established average-cost MDP literature. Given the remaining conceptual ambiguities and the absence of a clear discussion situating the contribution within existing theory, I believe the paper would benefit from further refinement before acceptance. As such, I lean toward rejection in its current form.

**Reviewer Concerns:**

### Addressed

**Sample-based algorithms**. Reviewers 2pSR and aaNs raised questions about whether the proposed policy gradient theorem can be operationalized, including whether a sample-based algorithm can be derived. The authors provided a sketch of such an approach in the rebuttal, which partially addresses these concerns.

**Clarifying questions and minor issues.**. Several reviewers raised requests for clarification regarding definitions, convergence claims, and specific examples. The rebuttal responded to many of these points and corrected minor ambiguities and presentation issues.

### Outstanding

**Connection to the average-cost MDP literature**. While some reviewers (F2er, aaNs) raised the connection to the average-cost MDP literature, the authors did not elaborate on this in a meaningful way. In the rebuttal, the authors state that they “elaborated on the distinction between the average-cost setup and our undiscounted total-reward setup in the related works section of the revised manuscript,” but the only added statement appears to be: “Our setup is also distinct from the average-reward setup, which considers averaging the sum of rewards.” This clarification is superficial and does not address the substantive theoretical relationship between the two settings.

**Limited technical novelty**. Almost all reviewers raised concerns about limited technical novelty. While Reviewer 2pSR viewed this as a minor issue, Reviewer aaNs explicitly stated that the novelty concern remains unresolved after the rebuttal, and other reviewers expressed negative views on this point. The rebuttal does not convincingly articulate what fundamentally new insights this enables.

**Reviewer Scores:**

* Reviewer mXRp (initial score: 6). The reviewer argues that the paper's technical novelty is limited, as the analysis largely mirrors prior work with only a change in state visitation measures, the policy gradient results are overshadowed by stronger existing NPG guarantees, and the reliance on finite state–action spaces limits the practical significance of the results. While the authors' rebuttal addressed most of the questions, concerns about technical novelty remain, so I would expect the reviewer to maintain their initial rating.

* Reviewer 2pSR (initial score: 8). The reviewer finds the paper's technical novelty limited and largely based on existing techniques, while raising concerns about unclear convergence-rate claims, the practical operability of the policy gradient theorem, and the need for more intuition and clarification around when and why linear convergence can occur. The authors' rebuttal addressed these questions and clarified the identified weaknesses. As a result, I would expect the reviewer to maintain their initial positive score.

* Reviewer F2er (initial score: 0). The reviewer argues that the paper neglects established average-cost MDP theory, relies on an unjustified and mathematically inconsistent finiteness assumption for undiscounted value functions, and therefore lacks a sound theoretical foundation. While the reviewer did not specify which prior results on average-cost MDPs are most relevant, the authors also did not clearly explain the connection between the two settings in the rebuttal. As a result, I would expect the reviewer to maintain their score or increase it only slightly after the discussion.

* Reviewer aaNs (initial score: 4). The reviewer argues that the paper relies on a strong and misleading finiteness assumption that effectively reduces the problem to a transient MDP with limited technical novelty, leaves learning implications unclear, and insufficiently positions the work relative to finite-horizon practice and the average-reward MDP literature. The rebuttal does not fully address these concerns, and the reviewer explicitly stated that they would maintain their original rating.

* Reviewer LNyM (initial score: 2). The reviewer finds the paper's theoretical analysis vague, internally inconsistent, and potentially ill-defined (especially around transient visitation and policy gradients), with limited novelty over prior work, and argues that the toy empirical results do not meaningfully validate or corroborate the theoretical claims. While the authors' rebuttal addressed many of the reviewer's questions in detail, some concerns remain unresolved, especially around the proposed pathologies. The Rock–Paper–Scissors example raised by the reviewer should not be interpreted as a two-player game since the opponent's policy is fixed. Overall, I would expect the reviewer to maintain their initial score or possibly increase it slightly to acknowledge the authors' clarifications and corrections of minor issues.

---

### Decision · Program_Chairs · 2026-01-26

Reject